# Implementation of Modern Therapeutic Drug Monitoring and Lipidomics Approaches in Clinical Practice: A Case Study with Colistin Treatment

**DOI:** 10.3390/ph17060753

**Published:** 2024-06-07

**Authors:** Ivana Gerhardtova, Ivana Cizmarova, Timotej Jankech, Dominika Olesova, Josef Jampilek, Vojtech Parrak, Kristina Nemergutova, Ladislav Sopko, Juraj Piestansky, Andrej Kovac

**Affiliations:** 1Institute of Neuroimmunology, Slovak Academy of Sciences, Dubravska cesta 9, 845 10 Bratislava, Slovakia; ivka.gerhardtova@gmail.com (I.G.); timotej.jankech@gmail.com (T.J.); dominika.olesova@savba.sk (D.O.); josef.jampilek@gmail.com (J.J.); vojtech.parrak@savba.sk (V.P.); 2Department of Analytical Chemistry, Faculty of Natural Sciences, Comenius University Bratislava, Ilkovicova 6, 842 15 Bratislava, Slovakia; 3Department of Pharmaceutical Analysis and Nuclear Pharmacy, Faculty of Pharmacy, Comenius University Bratislava, Odbojarov 10, 832 32 Bratislava, Slovakia; ivana.cizmarova@fpharm.uniba.sk; 4Toxicological and Antidoping Center, Faculty of Pharmacy, Comenius University Bratislava, Odbojarov 10, 832 32 Bratislava, Slovakia; 5Institute of Experimental Endocrinology, Biomedical Research Center SAS, Dubravska cesta 9, 845 10 Bratislava, Slovakia; 6Clinic of Hematology and Transfusiology, St. Cyril and Methodius Hospital, Antolska 11, 851 07 Bratislava, Slovakia; kika.nemergutova@gmail.com (K.N.); sopko.ladislav@gmail.com (L.S.); 7Department of Galenic Pharmacy, Faculty of Pharmacy, Comenius University Bratislava, Odbojarov 10, 832 32 Bratislava, Slovakia; 8Department of Pharmacology and Toxicology, University of Veterinary Medicine and Pharmacy in Kosice, Komenského 68/73, 041 81 Kosice, Slovakia

**Keywords:** therapeutic drug monitoring, colistin, critically ill patients, lipidomics, liquid chromatography, capillary electrophoresis, mass spectrometry

## Abstract

Nowadays, lipidomics plays a crucial role in the investigation of novel biomarkers of various diseases. Its implementation into the field of clinical analysis led to the identification of specific lipids and/or significant changes in their plasma levels in patients suffering from cancer, Alzheimer’s disease, sepsis, and many other diseases and pathological conditions. Profiling of lipids and determination of their plasma concentrations could also be helpful in the case of drug therapy management, especially in combination with therapeutic drug monitoring (TDM). Here, for the first time, a combined approach based on the TDM of colistin, a last-resort antibiotic, and lipidomic profiling is presented in a case study of a critically ill male patient suffering from *Pseudomonas aeruginosa*-induced pneumonia. Implementation of innovative analytical approaches for TDM (online combination of capillary electrophoresis with tandem mass spectrometry, CZE-MS/MS) and lipidomics (liquid chromatography–tandem mass spectrometry, LC-MS/MS) was demonstrated. The CZE-MS/MS strategy confirmed the chosen colistin drug dosing regimen, leading to stable colistin concentrations in plasma samples. The determined colistin concentrations in plasma samples reached the required minimal inhibitory concentration of 1 μg/mL. The complex lipidomics approach led to monitoring 545 lipids in collected patient plasma samples during and after the therapy. Some changes in specific individual lipids were in good agreement with previous lipidomics studies dealing with sepsis. The presented case study represents a good starting point for identifying particular individual lipids that could correlate with antimicrobial and inflammation therapeutic management.

## 1. Introduction

Antimicrobial resistance represents a severe threat to human health worldwide in the 21st century [1]. Particular difficulties are associated with antibiotic-resistant Gram-negative pathogens, such as *Escherichia coli*, *Klebsiella pneumoniae*, *Acinetobacter baumannii*, and *Pseudomonas aeruginosa* [1,2,3,4]. The rise of antibiotic resistance led to the repurposing of some old antibiotics (ATBs), such as polymyxins (including colistin), which have recently been used as the last therapeutic option for infections caused by the aforementioned Gram-negative bacteria [5,6]. Colistin (CST, or polymyxin E) is a cyclic lipopeptide with a narrow antibacterial spectrum. Generally, it has two forms: colistin A (CST A, or polymyxin E_1_) and colistin B (CST B, or polymyxin E_2_). Differences in the chemical structures of these two forms are shown in Appendix A. Although polymyxins entered clinical practice in the late 1950s, their mechanism of action is still not well understood [7,8]. Polymyxins are expected to target lipid A, a specific component of the lipopolysaccharide on the bacterial outer membrane [9].

Moreover, information on these ATBs’ physicochemical and pharmacological properties is limited, significantly affecting their appropriate and safe use in clinical practice, especially in critically ill patients [10]. In such cases, therapeutic drug monitoring (TDM) of ATBs seems to be a promising tool for setting optimal therapy management of critically ill patients, including optimizing dosage regimens, minimizing unwanted side effects, and preventing bacterial resistance. Here, it is necessary to mention that achieving the therapeutic effect of ATBs is challenging, e.g., in morbidly obese patients, in patients with catheters, or in increased renal and/or hepatic function [11]. The correct dosage of ATBs is imperative to ensure their adequate exposure [12].

In general, TDM represents a measurement of drug concentration in body fluids or tissues, which is performed to optimize the patient’s therapy outcome [13]. Typically, it is performed for drugs with a narrow therapeutic index, drugs with a well-defined relationship between concentration and effect, and drugs with large inter- or intra-individual distribution or clearance differences. A drug is a potential candidate for TDM if there is a large interindividual pharmacokinetic variability and if the therapeutic effect of the drug cannot be adequately and easily measured [14]. CST meets the mentioned criteria, and measurement of its concentration in critically ill patients is highly demanded by clinicians to control and optimize the drug dosing regimen. 

The TDM procedure is composed of three phases: (i) pre-analytical phase, (ii) analytical phase, and (iii) post-analytical phase (proper clinical interpretation of the measured data). The pre-analytical phase is accompanied by appropriate planning of the sampling, sample collection, and storage. Here, knowledge about the exact time of drug administration is critical for reliable interpretation of measured data. In general, samples are collected at a steady state [15]. The results from the laboratory should be available within a short period of time, in the optimal situation, before the administration of the next dose [16]. In case of uncertainty about the therapeutic efficacy of the drug, samples are collected just before the administration of the next dose [17].

The analytical phase usually consists of the implementation of chromatographic (especially high-performance liquid chromatography, HPLC) and immunologic methods. However, they have some limitations, including the lack of standardization of work procedures, long analysis time, high costs, and demands for the complex preparation and pretreatment of samples. The above-mentioned weakness of the convenient methods could be overcome by new developing technologies based on capillary zone electrophoresis (CZE) [18] or biosensors [19]. However, the implementation of such analytical approaches in the clinical environment is not yet fully established. CZE is a promising alternative method for the analysis of drugs and their metabolites in biological fluids. It offers several advantages in TDM, i.e., relatively fast analysis time, simple instrumentation, environmental friendliness, better resolution, high separation efficiency allowing multicomponent analysis, and low cost of analysis compared to HPLC (only a small amount of solvent is required, and capillaries are relatively inexpensive). The drawback of CZE is the lack of sensitivity and suboptimal detection limits. However, these problems can be solved by appropriate sample pretreatment and/or combination with selective and sensitive detection (e.g., mass spectrometry, MS). CZE methods are usually used only for newer drugs in TDM, as HPLC assays or immunoassays are available for older drugs [20]. The dominant position of HPLC methods in TDM was also observed in the case of CST determination, where the instrumental approaches based on the hyphenation of liquid chromatography (LC) with fluorescence (FLD) [21,22,23,24] and/or MS detection [25,26,27,28,29,30,31,32,33,34,35,36,37,38,39,40,41] are the most common. There is only one CZE-MS/MS method for TDM of CST described in the scientific literature, and this approach was developed by our research group [42].

Recent trends in clinical practice are oriented toward personalized medicine. Such an approach typically demands a fundamental understanding of the disease, identification of drug targets for therapy, and the discovery of relevant biomarkers of the disease or monitoring the effectiveness of drug treatment, leading to clinical follow-up in medical therapy [43,44,45,46]. Nowadays, there is a significant rise in various fields of systems biology, which enables us to fulfill the aforementioned demands of clinicians. Metabolomic and lipidomic strategies are dominant ones and are accompanied by the development of new drugs and the exploration of new potential biomarkers for various diseases [47,48,49]. A combination of these strategies with TDM (especially in the case of ATB therapy) could be a very powerful and multivariate tool for unequivocal monitoring of the whole therapeutic protocol focused on the individual demands of the patient. Recently, a detailed metabolomics study was performed on CST methanesulfonate-treated *Mycobacterium tuberculosis* [50], which identified 22 significantly changed molecules. These findings may be helpful to clarify the mechanism of action of CST. Moreover, another very recent study by Carfrea et al. [51] demonstrated a significant impact of inhibition of fatty acid synthesis on overcoming CST resistance. This fact also confirms the increasing interest of scientists in lipidomic approaches and their application in the clinical field [52]. 

Recently, the main focus has been on lipidomics. In general, lipids are vital biomolecules deeply involved in diverse cellular processes and pivotal roles, both in healthy and diseased states. They form integral components of cell membranes, barriers, energy reservoirs, and signaling pathways [53,54]. Dysregulation of lipid homeostasis can lead to a spectrum of disorders, including cardiovascular diseases, diabetes mellitus, neurodegenerative diseases such as Alzheimer’s disease, and inflammation [53,55]. Lipidomics, as a part of systems biology, aids in understanding lipid metabolism, identifying biomarkers, and discovering therapeutic targets for lipid-related diseases [56,57], which was documented, e.g., in the case of non-alcoholic fatty liver disease and non-alcoholic steatohepatitis, cancer, or inflammation or in antiviral therapies [58]. Moreover, lipid profile helps to monitor the administration of clinical treatment and diagnose human diseases [59]. Recently, lipid changes have often been evaluated in septic patients [60,61,62,63]. It was demonstrated that sepsis is usually associated with the downregulation of lysophosphatidylcholine (LPC) levels, increased plasma free fatty acid (FA) and ceramide (Cer) species levels, or changes in the metabolism of polyunsaturated FAs (PUFA) in plasma [64]. 

In addition to sepsis, there are only a few publications dealing with changes in the lipid profiles of patients with inflammation or bacterial infection [65,66,67,68]. Chen et al. [66] performed lipidomic and metabolomic analyses of plasma and urine samples of children affected by *Mycoplasma pneumoniae* infection. As a result, 163 lipid molecules and 104 metabolites were identified in plasma samples and 208 metabolites in urine samples. The most significant changes in lipid species were observed in lysophosphatidylethanolamines (LPEs), phosphatidylinositols (PIs), phosphatidylcholines (PCs), phosphatidylethanolamines (PEs), and triglycerides (TGs). 

A study by Mansell et al. [65] dealt with changes in the metabolomic and lipidomic profile of infants up to 12 months of age after overcoming infectious diseases and inflammation in order to determine the impact of these changes on the possible risk of cardiovascular and metabolic diseases in adulthood. In children under one year with more frequent occurrence of infectious diseases, these diseases were associated with adverse metabolomic and lipidomic profiles, such as increased levels of inflammatory markers, TGs, PEs, phenylalanine, apolipoprotein A1, and high-density lipoprotein (HDL) cholesterol and, on the other hand, downregulation of trihexosylceramides, dehydrocholesteryl esters, and plasmalogens.

Scott et al. [68] identified 26 phospholipid ions (phosphatidic acid (PA), PE, phosphatidylglycerol (PG), PI, phosphatidylserine (PS)) in mouse lungs infected by *Pseudomonas aeruginosa*, 11 ions being unique to the parenchyma, 6 unique to the airways, and 9 ions being detected in both.

All the aforementioned papers focused on monitoring lipid changes before treatment (as potential biomarkers) or after treatment. A direct connection between lipidomics and TDM was realized only in one study by Ahn et al. [67], where the lipid profile during chemotherapy treatment of tuberculosis was monitored [67]. 82 plasma samples were analyzed by ultra-high-performance liquid chromatography in combination with quadrupole-time-of-flight MS (UHPLC-QToF-MS). After chemotherapy treatment, 17 subclasses of lipids were significantly changed: levels of cholesteryl esters (CE), monoacylglycerols (MAG), and PCs increased, while TAG, SM, and ether-linked phosphatidylethanolamines (PE O-) levels decreased. At the same time, potential markers for TDM were identified, for example, CE (24:6), PC (42:6), PE (O-40:5), and dihexosylceramide Hex2Cer (34:2;2 O).

In our work, for the first time, a comprehensive therapy optimizing strategy based on the implementation of TDM of CST performed by an advanced CZE-MS approach combined/supported with a complex lipidomics approach was applied. This strategy is documented in a case study of a critically ill male patient with pneumonia. Such a combination of advanced analytical approaches showed information regarding the appropriate dosage regimen of the applied ATB to obtain the demanded therapeutic effect and also demonstrated changes in lipid levels during the therapy. This case study could serve as a good starting point for another, more detailed study focused on lipid biomarkers of the ATB therapy efficacy.

## 2. Results and Discussion

Recent trends in clinical practice are oriented toward personalized medicine, which takes into account inter-individual variations and so contributes to tailored therapeutic management of various diseases. One of the first strategies towards personalized medicine is therapeutic drug monitoring (TDM), which enables to change the drug dose regimen according to individual physiological and pathological conditions. Similarly, “omics” represent tools that are very often used in the clinical environment to help to tailor the therapeutic regimen according to the unique biochemical, physiological, environmental exposure, and behavioral profile of the individual [69]. In our study, we present the implementation of two approaches in the clinical field of bacterial infections and their treatment. The first one is focused on TDM of CST, a lipopeptide ATB, realized using an innovative, highly effective, and sensitive method based on a combination of capillary zone electrophoresis (CZE) with tandem mass spectrometry (MS/MS). The second one is based on lipidomics involving precise identification and quantification of lipids by modern methods based on a combination of ultra-high-performance liquid chromatography (UHPLC) and MS/MS. These two approaches were used to analyze samples obtained from a critically ill patient on CST antibacterial therapy.

### 2.1. Case Presentation

A 38-year-old male patient suffering from acute myeloid leukemia (AML) was hospitalized with pneumonia caused by *Pseudomonas aeruginosa*. According to the standard therapeutic protocol, he was treated with the combination of two ATBs, i.e., meropenem (1 g every 8 h) and linezolid (600 mg every 12 h). According to the additional testing of the patient’s smears, positive cultivation of *Micrococcus luteus* was detected. In general, *M. luteus* is a harmless Gram-positive bacterium normally colonizing the human mouth, mucosae, oropharynx, and upper respiratory tract. However, if the patient is immunocompromised, it can become an opportunistic pathogen [70]. This was also the case of the hospitalized patient, and therefore, ATB gentamicin (80 mg each 8 h) was added to the drug management. Furthermore, secondary diseases were treated by amphotericin B (400 mg every 24 h). After 8 days of gentamicin therapy and 4 days of meropenem + linezolid therapy, the patient still showed extremely elevated C-reactive protein (CRP) values (335 mg/L), which indicated complicated bacterial infection. Because of the critical conditions of the patient, a last resort ATB, CST, was added to the therapeutic protocol. A loading dose of 9 million international units (MIU) of CST followed by a maintenance dose of 3 MIU every 8 h was used. CST was applied for 7 days. After three days of CST therapy, meropenem and linezolid were replaced by a fixed combination of avibactam with ceftazidime (2 g/0.5 g), which was administered every 8 h for 6 days. 

Blood samples were collected every day in the early morning before the administration of the drugs, except the first day when CST was applied; on this day, a blood sample was collected right before the application of the second dose. The required biochemical blood tests were carried out after the collection of the samples. An overview of some important parameters is provided in Table 1.

### 2.2. Therapeutic Drug Monitoring of CST

From the clinical point of view, CST has an extremely narrow therapeutic index, and the plasma concentration required for antibacterial activity can be similar to that which predisposes it to nephrotoxicity [71,72]. According to Regenthal et al. [73], the therapeutic plasma concentration range is 1–5 μg/mL. Therefore, the implementation of TDM is essential in such cases. As mentioned previously, blood samples were collected every morning before the intravenous administration of the drugs. This sampling procedure was in accordance with the TDM guidelines [17]. This sampling strategy was chosen due to the severe medical condition of the patient, but the optimal sampling time was not respected.

For TDM purposes, the total CST concentration calculated as a sum of CST A and CST B is required. Our previously developed and validated capillary zone electrophoresis-tandem mass spectrometry (CZE-MS/MS) method [42] was applied for the TDM of CST. The recent study should also prove the real application potential of the previously developed CZE-MS/MS approach because, in the pilot study, only one clinical sample containing CST was investigated. Here, the method was applied during the whole period of therapy to monitor CST levels in plasma, and when necessary, it should be a rational tool for drug dosage change. However, according to the determined levels of CST in plasma samples and clinical outcomes (significant decrease of CRP), a change in the dose regimen was not necessary. Relevant concentration levels determined by the proposed method during the whole therapy are summarized in Figure 1.

According to the results, our proposed analytical method was able to determine CST in each of the analyzed samples. The determined concentration on the first day of CST administration is affected by the time of sample collection. In this case, the blood sample was collected right before the application of the second dose (i.e., 3 MIU). Therefore, the determined concentration was very low (0.19 μg/mL) and does not represent a sufficient drug level required for the antibacterial effect. Further sample collection was standardized and realized every morning right before the application of the next CST dose.

As can be seen from Figure 1, the concentration of CST determined by CZE-MS/MS was maintained above the minimum inhibitory concentration of 1 μg/mL [73]. The proposed dosing regimen and further ATB therapy showed a good effect on the patient’s health, which was also confirmed by the decreasing value of CRP (see Table 1). 

The obtained results confirmed the possibility of using analytical approaches based on CZE in the specific field of TDM. The implemented CZE-MS/MS approach showed strengths such as simple sample pretreatment, minimal requirement on the sample amount, high separation efficiency, no carry-over effect, unequivocal identification, and quantification properties. These benefits are documented by illustrative records obtained from the analysis of the patient’s samples (Figure 2). 

### 2.3. Lipidomics

The lipidomics workflow was composed of two steps, including sample preparation and analysis. For simultaneous protein precipitation and lipid extraction, 2-propanol (IPA) was used. This procedure is very advantageous from a practical point of view, as it increases the throughput of the method by combining two sample preparation steps into one. The advantage of using IPA is that it can extract a relatively wide range of lipids. Moreover, IPA can be considered a greener solvent compared to other solvents (such as chloroform, methanol, or methyl *tert*-butyl ether—MTBE) used for lipid extraction. After extraction and subsequent centrifugation, the prepared sample was analyzed by the LC-MS/MS method, according to Sedlák et al. [74], which included 892 lipids in total, while the MRM transition was set for each lipid to simplify identification. We used a Hypersil GOLD C8 column for lipid separation. Compared to the published data that were acquired on the ACQUITY BEH C8 column [74], we achieved similar or even better separation of the analytes. Due to the column change, adjusting the expected retention times (t_R_) of all lipids was necessary to set the optimal retention time window. To verify the quality of the analysis, pooled QC samples were analyzed after every three injections of samples. A characteristic LC-MS/MS TIC chromatogram of the separation of all lipids in the patient’s plasma sample is shown in Figure 3A. An example of LC-MS/MS chromatograms of selected triglyceride, sphingomyelin, and fatty acid is present in Figure 3B–D. Using this method, we were able to identify more than 500 lipids in a relatively short analysis time (20 min). As previously demonstrated, LC-MS/MS is the method of choice not only for untargeted lipidomics but also for a targeted approach due to the number of analytes that can be determined in one run.

Finally, the modified lipidomic analytical method was applied to the samples from the critically ill patient. According to the proposed approach, we were able to monitor 545 individual lipids during the whole investigated period of time. The implemented method enabled investigation of the following lipid classes: cholesteryl esters (CE), ceramides (Cer), cholesterol (Chol), diglycerides (DG), fatty acids (FA), dihexosyl-ceramides (Hex2Cer), hexosyl-ceramides (HexCer), lysophosphatidylcholines (LPC), plasmenyl lysophosphatidylcholines (LPCO), lysophosphatidylethanolamines (LPE), plasmenyl lysophosphatidylethanolamines (LPEO), phosphatidylcholines (PC), plasmenyl phosphatidylcholines (PCO), phosphatidylethanolamines (PE), plasmenyl phosphatidylethanolamines (PEO), phosphatidylglycerols (PG), phosphatidylinositols (PI), phosphatidylethanolserines (PS), sphingomyelins (SM), and triglycerides (TG). 

The continuous plot of lipid classes, where the orange line shows the average concentration level of lipid classes determined after the follow-up visit realized approximately two months after the end of CST treatment and discharge of the patient from the hospital, is shown in Figure 4. There was a decrease in DG, FA, HexCer, LPC, LPCO, LPE, LPEO, PC, PCO, TG, PEO, and PI levels at the beginning of CST treatment. PS, PG, and Cer levels were slightly increased, and CE, SM, Chol, Hex2Cer, and PE remained unchanged. Interestingly, we observed a drop in the concentration of all lipid classes at day 3 after the start of CST treatment. This correlates with the elevation of the ALT and AST enzymes on the same day. The elevated levels of ALT and AST can often be indicators of temporary liver injury or damage, which is common with antibiotic use [75]. Liver dysfunction associated with an increase in ALT and AST can indirectly affect lipid metabolism, for example, in lipid synthesis or storage. The same trend can be observed on days 5 and 6 of CST treatment when the ALT and AST levels were again slightly elevated, while the levels of some lipids were decreased again. Some lipid classes, including FA, PC, LPC, PS, and others, play roles in the host defense against bacterial infection and serve as energy reservoirs [76,77]. During infection, the immune system becomes activated and requires energy and resources to mount a defense response. Thus, these lipid classes may be consumed or converted into other forms during the infection process, which could cause a decrease in their levels. As can be seen in Figure 4, the levels of such lipids as FA, PC, LPC, LPCO, LPE, and PS changed during the treatment, likely due to their involvement in the immune response to the infection. After overcoming the infection and stabilizing the patient, their levels increased. In general practice, the concentration profiles of lipids could be affected by the diet regime. However, in our study of the critically ill patient, standardized nutrition was administered intravenously. 

After the positive effect of CST, other ATBs administered showed a synergistic effect with CST, and such a therapeutic combination was able to efficiently treat the patient. As the CRP decreased and other clinical parameters stabilized, the concentrations of almost all lipid classes increased (Figure 5). 

The above data applies to the results of the entire lipid classes and does not reflect changes in individual lipids in the given class. Therefore, it was necessary to find specific lipids whose levels were changed during critical bacterial infection and under septic conditions (CRP > 400 mg/L) that were present in this patient (Appendix A). 

There is a lack of studies dealing with lipid changes in conjunction with bacterial infections. However, in the case of sepsis, some lipidomic studies were performed, and significant changes in individual lipids were observed. A lipidomic study of plasma and erythrocytes carried out by Mecatti et al. [64] showed lipid changes in septic and control patients. In this research, the downregulation of LPCs and sphingomyelin (SM) with specific FA chains and the upregulation of phosphatidylcholines (PCs) were observed. The results obtained by our study showed seven SMs, nine LPCs, and four PCs, and they are in good correlation with this study. The concentrations of all SMs and LPCs also decreased during our study due to ongoing bacterial infection. Regarding PCs, the change of two lipids—PC (16:0/18:2) and PC (16:0/20:1)—was the same. Two lipids, PC (16:0/20:5) and (16:0/20:3), showed discrepancies in comparison with the findings by Mecatti et al. [64]. These discrepancies may be caused by the use of different protocols applied for the extraction of the lipids. Moreover, the study by Mecatti et al. includes a large number of probands and healthy control subjects. 

Similarly, Sulaiman et al. [60] found that the levels of three free FAs and four lipids (11-hydroxyeicosatetraenoic acid (11-HETE), 12-HETE, 15-HETE, and 14(S)-hydroxy docosahexaenoic acid) from arachidonic acid’s pathway were significantly changed in rapid recovery patients compared to chronic critical illness (CCI) or early death patients. The level of FA 12:0 was decreased in CCI/early death patients, while levels of FA 17:0 and FA 20:1 were increased. Moreover, all four lipids mentioned above were significantly increased in CCI/early death patients. 

In another study, the lipidomic profile of 60 patients with bacterial sepsis (Sepsis-2 and Sepsis-3) was investigated. Samples from patients were analyzed before starting antibiotics and supportive treatment. The difference between the Sepsis-3 and Sepsis-2 patient groups was mostly in SM metabolism. Patients with or without bacteremia showed differences in PC and triacylglycerols (TAGs). Sepsis-3 patients differed in their metabolic profile depending on the severity of their disease [62]. Only FA 20:1 was decreased with an ongoing bacterial infection in our study.

A study of patients with celiac sepsis observed significant changes in lipid profiles. Unfortunately, the authors did not specify antimicrobial therapy. Sixty-four lipid levels from 65 identified molecules were decreased in septic patients, while the level of PE (34:2) was increased compared to that in control group patients. All major PCs were decreased [63]. 

### 2.4. TDM of CST in View of Lipidomics Data

In this work, the combination of TDM with complex lipidomic profiling before, during, and after the administration of CST was realized for the first time. As mentioned in Section 2.3, which focused on the results of lipidomics experiments, a drop in concentrations of all lipid classes on day 3 after CST administration was observed. This observation could probably be associated with CST administration. It is well known that the steady-state concentration of CST is typically obtained after three days [78] and reaching this pharmacokinetic parameter could be accompanied by changes in the concentration of some substances in biological fluids. However, this suggestion could not be confirmed in our study due to the limitations, such as the single-patient design of the study and the absence of untreated control samples. Further changes in the lipidomic profile and concentration levels of the investigated samples could also be associated with the concomitant ATB modalities used in the therapeutic protocol. To confirm and clearly evaluate potential lipidomic markers of CST therapy efficacy, it would be necessary to carry out a properly designed study that includes a higher number of probands and untreated control subjects. Recent results are able to present only a limited view of the association of CST plasma levels with changes in the lipid profile. 

### 2.5. Drawbacks of the Case Study and Future Perspectives

The presented study highlighted the strong analytical potential of the developed methods for TDM of CST and lipidomics in critically ill patients. The main drawback of the lipidomic analysis is that only one critically ill patient was involved in the study. The single-patient design of the study was not confronted and compared with untreated control samples. Therefore, the data cannot be generalized to all patients suffering from microbial infections. To improve the clinical outcome of the study, it will be necessary to monitor a large group of patients on CST therapy. Moreover, accurate stratification of patients into specific subgroups (e.g., responders vs. non-responders to the ATB treatment) will be of great importance. Such a study design that combines TDM and lipidomics will be able to identify and confirm specific individual lipid molecules that serve as markers for successful ATB treatment of critically ill patients. We expect that in the near future, the incorporation of lipidomic approaches into the therapeutic protocols of bacterial infections, inflammation, and sepsis, together with TDM of some last-resort ATBs (including CST) used in the therapy of these diseases, will lead to improvements towards personalized medicine. Moreover, such multivariate data obtained during therapy will be associated with the possibility of an early intervention into therapy and offer better prediction of the cure prognosis. Similarly, we expect that the combination of TDM with lipidomics will significantly influence antibiotic stewardship and so the responsible use of ATBs, with the goal of preserving their future effectiveness and protecting public health. 

## 3. Materials and Methods

### 3.1. Therapeutic Drug Monitoring of CST

#### 3.1.1. Chemicals and Samples

Formic acid (LC-MS grade), methanol (LC-MS grade), and acetonitrile (LC-MS grade) were obtained from Merck (Darmstadt, Germany). Sodium hydroxide (p.a. quality) was purchased from Agilent Technologies (Santa Clara, CA, USA). The ultrapure water used for the preparation of the background electrolyte (BGE), sheath liquid, and samples was produced by Millipore Direct-Q^®^ UV Water Purification System (Merck). Analytical-grade standards of colistin sulfate and polymyxin B sulfate were obtained from Sigma Aldrich (Steinheim, Germany).

#### 3.1.2. Instrumentation

The CZE-MS/MS experiments were realized with the use of an Agilent 7100 capillary electrophoresis system (Agilent Technologies) online connected with an Agilent 6410 Series Triple Quadrupole (Agilent Technologies) tandem mass spectrometer equipped with a commercial coaxial sheath liquid electrospray (ESI) interface. The CZE separation procedure was performed in a 99 cm × 50 μm inner diameter (ID) bare fused silica capillary (MicroSolv Technology Corporation, Eatontown, NJ, USA). The separation procedure was performed in a background electrolyte (BGE) composed of 50 mM formic acid. Hydrodynamic injection of the samples was performed at the pressure of 50 mbar for 20 s, followed by an injection of BGE for 2 s at 50 mbar. An internal pressure of 5 mbar was applied from the second minute of the electrophoretic analysis. 

CZE-MS connected via the ESI interface demanded delivery of sheath liquid composed of methanol/0.1% formic acid (50/50, *v*/*v*) solution at a flow rate of 8 μL/min. The tandem mass spectrometer operated in positive ion multiple reaction monitoring (MRM) mode and following precursor ion–product ion transitions were applied: CST A 390.7 → 101.1 (quantification transition), 390.7 → 211.6 (qualification transition); CST B 386.0 → 101.1 (quantification transition), 386.0 → 326.9 (qualification transition); polymyxin B (used as internal standard) 402.1 → 101.0 (quantification transition), 402.1 → 378.4 (qualification transition). The dwell time was 150 ms, capillary voltage 5000 V, nebulizing gas pressure 5 psi, drying gas temperature 300 °C, and drying gas flow 5 L/min.

#### 3.1.3. Preparation of Standards and Samples

The stock solutions of CST reference standard and its internal standard, i.e., polymyxin B, were prepared by dissolving 1 mg of their powder in 1 mL of ultrapure water. The solutions were aliquoted and kept at −80 °C. 

Plasma samples from one critically ill patient were collected over 10 days at the Department of Hematology and Transfusiology of St. Cyril and Methodius Hospital in Bratislava. After collection, the samples were stored at −80 °C. Before the analysis, the samples were prepared according to the procedure described in our previous publication [42]. The whole sample pretreatment demanded only a simple protein precipitation procedure using acetonitrile with 0.1% formic acid.

### 3.2. Lipidomics

#### 3.2.1. Chemicals and Samples

Ammonium acetate (NH_4_Ac) was purchased from Sigma-Aldrich (St. Louis, MO, USA), a mixture of lipid internal standards (IS) SPLASH^®^ Lipidomix^®^ was obtained from Avanti Polar Lipids (Alabaster, AL, USA). 2-propanol (IPA) and water (MPW) (LC-MS grade) were purchased from Supelco (Bellefonte, PA, USA). LC-MS grade acetonitrile (ACN) for mobile phase (MP) was purchased from Honeywell (Charlotte, NC, USA).

#### 3.2.2. Instrumentation

Lipid analysis was conducted using a targeted lipidomic approach on an Acquity LC system (Waters, Milford, MA, USA) coupled with tandem mass spectrometry (QTRAP^®^ 4500, AB Sciex, Framingham, MA, USA). Chromatographic separation was performed on a Hypersil GOLD C8 column (2.1 × 100 mm, 1.9 µm, Thermo Fisher Scientific, Waltham, MA, USA) with a pre-column Acquity C18 (1.7 µm, Waters). The mobile phase A (MP A) comprised 10 mM NH_4_Ac in a mixture of ACN:MPW in a ratio of 60:40 (*v*/*v*), while the mobile phase B (MP B) consisted of 10 mM NH_4_Ac in a mixture of IPA:ACN in a ratio of 90:10 (*v*/*v*), flowing at a rate of 0.35 mL/min. The column temperature was maintained at 55 °C, and the autosampler temperature was set to 4 °C. A gradient elution (shown in Table 2) was used for separation, with a total run time of 20 min.

Mass spectra were acquired using scheduled multiple reaction monitoring (MRM) in both positive and negative ionization modes. Capillary voltages in the mass spectrometer were set to +4500 V for positive ionization mode and −4500 V for negative ionization mode, with the source temperature set to 500 °C and the curtain gas pressure at 40 psi.

#### 3.2.3. Preparation of Quality Control (QC) Samples and Clinical Samples

For lipid extraction, samples were prepared by mixing 20 µL of human plasma with 60 µL of IPA and spiked with 3% of IS (2.4 μL). The mixture was vortexed for 30 s and put in a freezer for 30 min (−80 °C) for deproteinization. After defrosting, samples were centrifugated (10 min, 17,500 rpm, 4 °C), and the supernatant was transferred into vials prior to the analysis.

The pooled QC sample was prepared as follows: 10 µL of each sample supernatant was transferred into the tube, mixed, and centrifugated (4 °C, 17,500 rpm, 10 min). An aliquot of the pooled QC sample was transferred into a vial for analysis.

#### 3.2.4. Data and Statistical Analysis

The data were acquired using Analyst Software ver. 1.6.2, and MultiQuant 2.1 (AB Sciex) software was used to identify lipids. All statistical analyses and data visualization were performed using R software (version 3.5.0) and the Metabol function package [79].

## 4. Conclusions

This research focused on the implementation of innovative analytical tools based on CZE-MS/MS and LC-MS/MS approaches in the field of TDM and lipidomics with their perspective implementation into clinical practice. Combining these two modern approaches led to the determination of the appropriate dosing regimen for the ATB drug colistin and enabled the monitoring of concentration level changes of 20 lipid classes in plasma samples of the critically ill patient treated with colistin and other ATB drugs. The lipidomic approach allowed the analysis of 545 individual lipids. Furthermore, compared to previously published studies, the presence and/or concentration changes of some individual lipids could be a marker of the septic condition in the patient. However, additional clinical studies investigating large groups of patients and their appropriate subgroup stratification are needed to verify these trends in lipid changes. We expect that the combination of TDM with lipidomics will significantly influence personalized ATB treatment and antibiotic stewardship.

## Figures and Tables

**Figure 1 pharmaceuticals-17-00753-f001:**
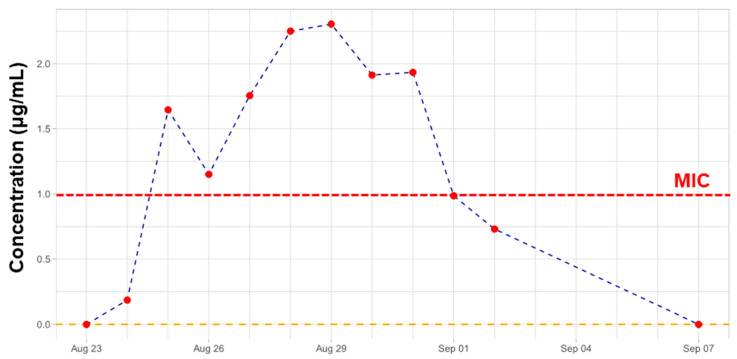
TDM of colistin. Overview of concentration levels of colistin in plasma samples obtained from the critically ill patient on colistin therapy. MIC—minimal inhibitory concentration.

**Figure 2 pharmaceuticals-17-00753-f002:**
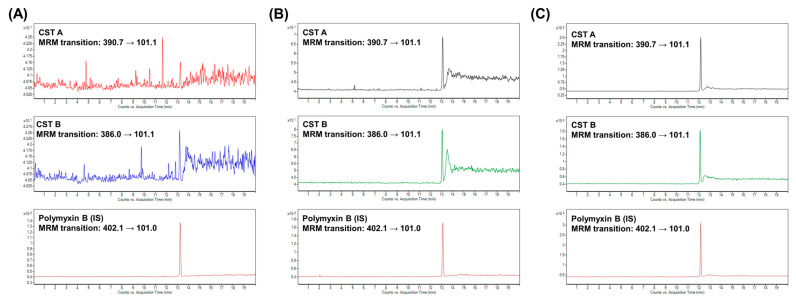
Illustrative multiple reaction monitoring (MRM) records of selected quantitative transitions for CST A, CST B, and their internal standard (IS), polymyxin B, obtained from the CZE-MS/MS analysis of colistin in the samples from the monitored critically ill patient. (**A**) Analysis of plasma samples from the patient before the treatment with colistin (Day 1 in Table 1). (**B**) Analysis of a plasma sample collected right before the first application of colistin at 3 MIU dose (Day 2 in Table 1). (**C**) Analysis of a plasma sample collected at day 6 of colistin treatment (Day 7 in Table 1).

**Figure 3 pharmaceuticals-17-00753-f003:**
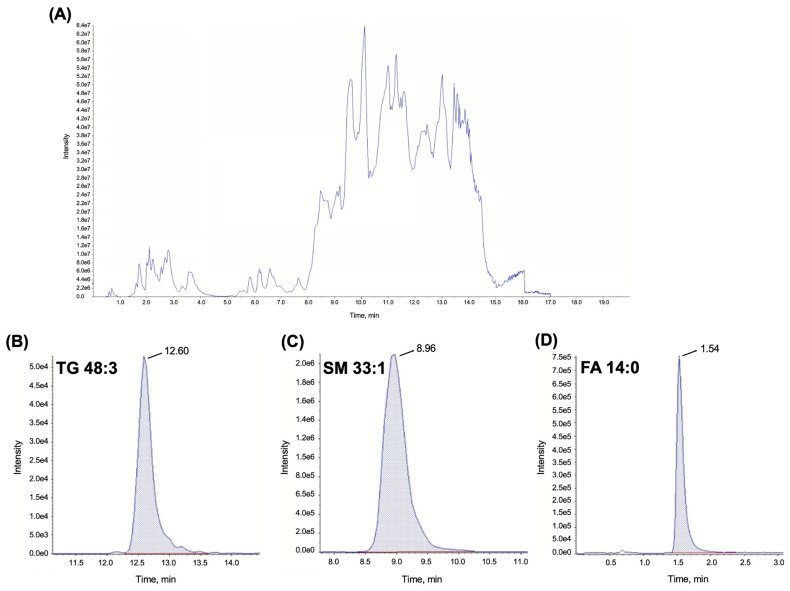
Lipidomics analysis of critically ill patient’s plasma sample collected on day 6 of colistin therapy. (**A**) Total ion current (TIC) chromatogram representing all more than 800 investigated individual lipids. (**B**) Extracted ion chromatogram (EIC) for individual lipid TG 48:3. (**C**) EIC for individual lipid SM 33:1. (**D**) EIC for individual lipid FA 14:0.

**Figure 4 pharmaceuticals-17-00753-f004:**
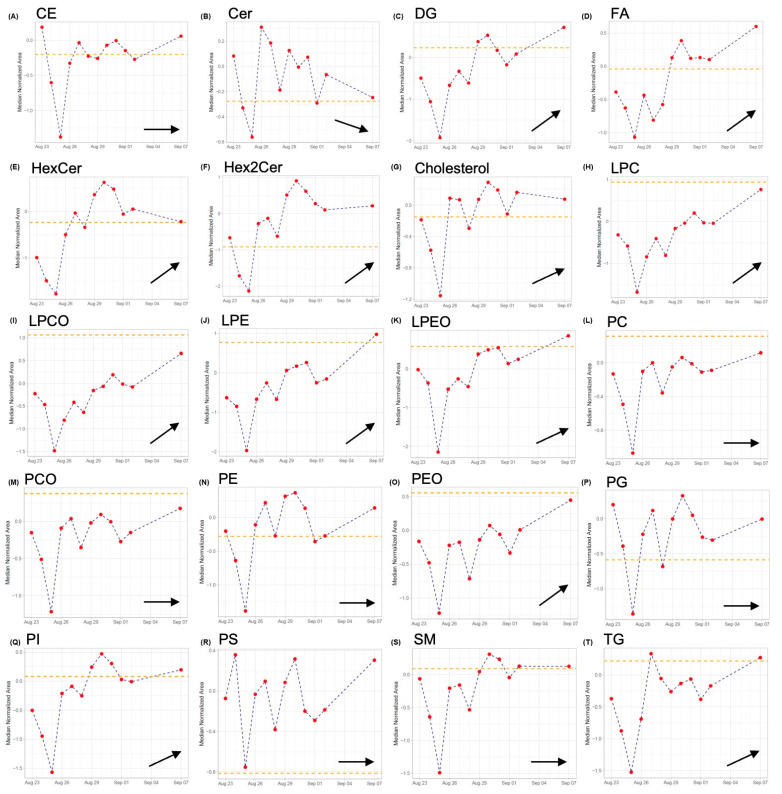
A complex overview of concentration changes in investigated lipid classes during the hospitalization of the critically ill patient and his treatment with multiple ATBs, including colistin. The orange dot line in each subgraph represents the average concentration level of the relevant lipid class determined after the follow-up examination. Changes in concentrations of (**A**) CE—cholesteryl esters; (**B**) Cer—ceramides; (**C**) DG—diacylglycerols; (**D**) FA—fatty acids; (**E**) HexCer—hexosyl-ceramides; (**F**) Hex2Cer—dihexosyl-ceramides; (**G**) Chol—cholesterol; (**H**) LPC—lysophosphatidylcholines; (**I**) LPCO—plasmenyl lysophosphatidylcholines; (**J**) LPE—lysophosphatidylethanolamines; (**K**) LPEO—plasmenyl lysophosphatidylethanolamines; (**L**) PC—phosphatidylcholines); (**M**) PCO—plasmenyl phosphatidylcholines; (**N**) PE—phosphatidylethanolamines; (**O**) PEO—plasmenyl phosphatidylethanolamines; (**P**) PG—phosphatidylglycerol; (**Q**) PI—phosphatidylinositols; (**R**) PS—phosphatidylethanolserines; (**S**) SM—sphingomyelins; and (**T**) TG—triacylglycerols.

**Figure 5 pharmaceuticals-17-00753-f005:**
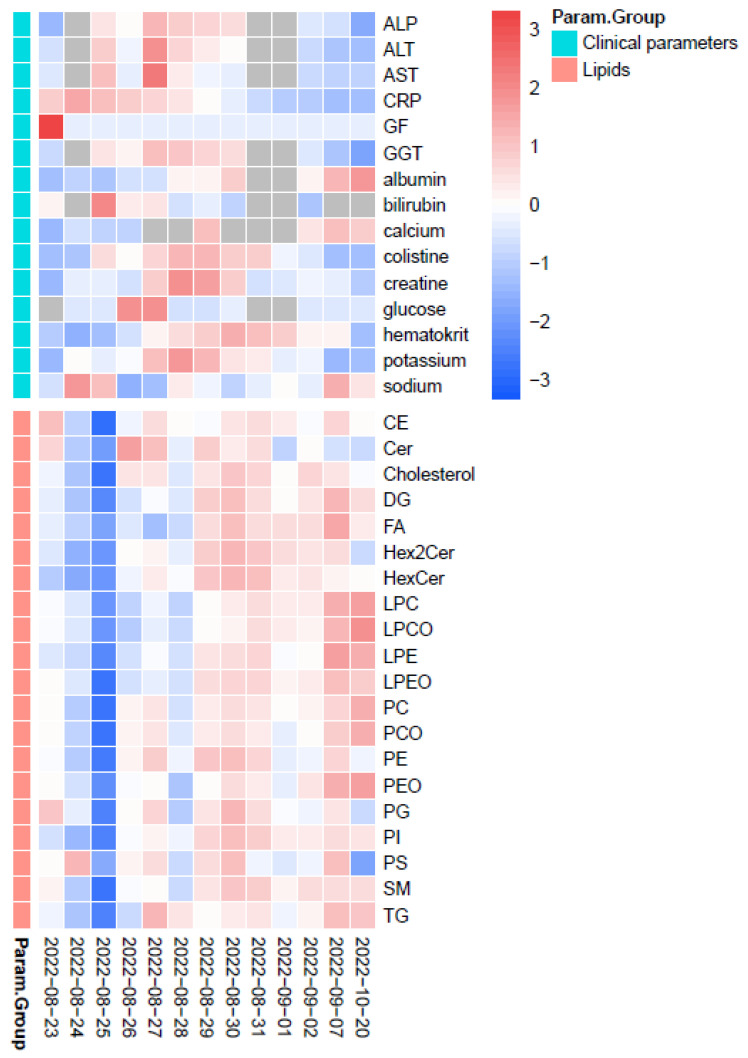
Heatmap of changes in lipid classes based on lipidomics and changes of clinical parameters on different days of patient monitoring. The dark blue color shows the lowest level of individual parameters/lipids, while the darkest red indicates the highest level of the given parameters. Abbreviations: ALP—alkaline phosphatase; ALT—alanine transaminase; AST—aspartate transaminase; CRP—C-reactive protein; GF—glomerular filtration; GGT (gamma-glutamyl transferase).

**Table 1 pharmaceuticals-17-00753-t001:** Overview of routinely determined biochemical parameters during the hospitalization and follow-up of the patient.

Clinical Parameter	Day 1	Day 2 ^*1^	Day 3	Day 4	Day 5	Day 6	Day 7	Day 8 ^*2^	Day 9	Day 10	Day 11	Day 12	Follow Up ^*3^
Serum creatinine (μmol/L)	70.6	89.9	89.7	83.9	109.7	129	125	111	84	85.7	92.3	89.5	76.2
GF (mL/s)	1.89	1.56	1.56	1.69	1.22	1.00	1.04	1.20	1.69	1.65	1.51	1.56	1.83
ALT (μkat/L)	1.27	n.d.	3.46	2.00	5.00	3.24	2.66	2.37	n.d.	n.d.	1.12	0.53	0.35
AST (μkat/L)	1.46	n.d.	5.27	1.87	8.34	3.40	2.21	1.83	n.d.	n.d.	0.78	0.57	0.43
GGT (μkat/L)	3.79	n.d.	7.10	6.31	9.11	8.66	8.04	7.61	n.d.	n.d.	4.47	2.78	0.77
ALP (μkat/L)	2.28	n.d.	5.14	4.37	6.32	5.60	5.55	5.30	n.d.	n.d.	3.66	3.56	1.89
Albumin (g/L)	26.6	28.6	27.0	29.5	29.6	33.0	33.0	35.9	n.d.	n.d.	33.0	38.0	40.7
CRP (mg/L)	335	442	364.8	338	313.4	263.5	201.0	153.4	80.2	49.5	37.26	8.35	5.37
Glucose (mmol/L)	n.d.	5.88	5.77	16.19	16.4	4.9	5.18	6.11	n.d.	n.d.	5.32	5.82	5.58
Bilirubin (mmol/L)	8.4	n.d.	11.8	8.6	8.8	6.9	7.4	6.5	n.d.	n.d.	6.1	n.d.	n.d.
Na (mmol/L)	134	145	142	130	131	138	136	133	135	137	135	143	139
K (mmol/L)	3.41	3.94	3.80	3.92	4.38	4.60	4.40	4.10	4.06	3.80	3.88	3.40	3.48
Hematocrit (%)	26.4	24.4	25.1	27.3	29.6	30.7	32.3	34	32.8	32.4	30	30	25.3

Abbreviations: GF—glomerular filtration; ALT—alanine transaminase; AST—aspartate transaminase; GGT—gamma-glutamyltransferase; ALP—alkaline phosphatase; CRP—C-reactive protein; Na—sodium; K—potassium; n.d.—not determined. ^*1^ The day when colistin was first administered. ^*2^ The day when colistin was last administered. ^*3^ Biochemical parameters investigated after the follow-up examination of the patient on an outpatient basis 51 days after hospitalization.

**Table 2 pharmaceuticals-17-00753-t002:** Gradient elution for lipid separation.

Time (min)	MP A [%]	MP B [%]
0.00	68	32
1.50	68	32
15.50	15	85
15.60	3	97
18.00	3	97
18.01	68	32
20.00	68	32

## Data Availability

The data presented in this study are available on request from the corresponding authors (privacy reason).

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
