# Peer review of "Implementation of Modern Therapeutic Drug Monitoring and Lipidomics Approaches in Clinical Practice: A Case Study with Colistin Treatment"

_pharmaceuticals, 2024, doi:10.3390/ph17060753_

Round 1

Reviewer 1 Report

Comments and Suggestions for Authors

This article presents an interesting case study on the application of therapeutic drug monitoring (TDM) for the antibiotic colistin using capillary electrophoresis-mass spectrometry (CZE-MS/MS) combined with lipidomic profiling in a critically ill patient with a bacterial infection. This innovative approach has the potential to provide new insights into how to monitor and optimize antibiotic therapy in critically ill patients. Several things need to be considered for the perfection of this manuscript:

·         The rationale for selecting colistin as the antibiotic to focus on is not fully explained. Providing more details on why monitoring colistin levels and lipidomic changes during colistin treatment is of particular interest would strengthen the introduction.

·         The introduction could benefit from a clearer description of the current gaps in knowledge that this study aims to address. What specific questions about using TDM and lipidomics to manage antibiotic therapy in critically ill patients does this work attempt to answer?

·         The lipidomic analysis methods are described in detail, but there is limited information provided about the validation and quality control measures used to ensure data reliability, such as the use of internal standards, calibration curves, and replicate analyses.

·         The data processing and statistical analysis methods for the lipidomic results are not fully explained. More details on how raw data were handled, normalized, and analyzed would improve transparency and reproducibility.

·         The generalizability and clinical relevance of the findings are not discussed. As this is a single-patient case study, it is important to consider the limitations of extrapolating the results to other patients and settings, and to frame the conclusions accordingly.

·         The potential impact of the findings on clinical practice is not addressed. Discussing how the TDM and lipidomic approaches could be implemented in real-world settings and what barriers need to be overcome would enhance the translational relevance of the conclusions.

·         The results and discussion do not draw clear connections between the TDM and lipidomic findings. Integrating these two aspects more directly could yield insights into how colistin levels might influence lipid profiles and vice versa.

·         Comparisons to previous lipidomic studies in sepsis are made, but the discussion could benefit from more explicit consideration of how the current findings align with or diverge from prior work, and what new knowledge this study contributes.

·         Potential limitations of the study, such as the single-patient design and lack of untreated control samples, are not addressed. Acknowledging and discussing these constraints would provide important context for interpreting the results.

Author Response

Dear Editors,

Dear Referees,

At the beginning, I would like to thank you for your interest in our manuscript (manuscript ID: pharmaceuticals-3027108), “Implementation of modern therapeutic drug monitoring and lipidomics approaches in clinical practice: A case study with colistin treatment” (authors Gerhardtova,I., Cizmarova,I., Jankech,T., Olesova,D., Jampilek,J., Parrak,V., Nemergutova,K., Sopko,L., Piestansky,J.*, Kovac,A.*), which has been submitted to your Journal (Pharmaceuticals) – special issue: "Innovative Tools for Drug Analysis and Therapeutic Drug Monitoring (TDM)".

On 17 May 2024 I received the reports to the manuscript. Please find my point-by-point reply to the comments and to the proposed amendments attached to this message. The paper was modified following the recommendations of the referees. The changes that have been made in the text are clearly marked. The "Response to Reviews" is included to this submission.

Finally, I would like to express my thanks to you and to the referees for the kind handling, valuable advice and suggestions on the manuscript. I look forward to further cooperation.

Sincerely Yours,

Juraj Piestansky & Andrej Kovac

(Corresponding author)

Bratislava, May 30, 2024

Assoc. prof. Juraj Piestansky

Department of Galenic Pharmacy,

Faculty of Pharmacy,

Comenius University,

Odbojarov 10,

SK-832 32 Bratislava,

Slovak Republic

Phone: **421-2-50 117 250

E-mail: piestansky@fpharm.uniba.sk

Dr. Andrej Kovac

Institute of Neuroimmunology,

Slovak Academy of Sciences,

Dubravska cesta 9,

845 10 Bratislava,

Slovak Republic

E-mail: andrej.kovac@savba.sk

Reviewer #1:

Statement of the Reviewer 1:

This article presents an interesting case study on the application of therapeutic drug monitoring (TDM) for the antibiotic colistin using capillary electrophoresis-mass spectrometry (CZE-MS/MS) combined with lipidomic profiling in a critically ill patient with a bacterial infection. This innovative approach has the potential to provide new insights into how to monitor and optimize antibiotic therapy in critically ill patients. Several things need to be considered for the perfection of this manuscript:

The rationale for selecting colistin as the antibiotic to focus on is not fully explained. Providing more details on why monitoring colistin levels and lipidomic changes during colistin treatment is of particular interest would strengthen the introduction.

Reply to the comments of the Reviewer 1:

Thank you for your recommendation. Recently, there is a high scientific interest in the identification and clarification of the mechanism of action of colistin and also in understanding and discovering the mechanism of antibiotics resistance. These are the main goals why the drug colistin was selected. It is an old ATB, but its mechanism of action is still not well understood. Moreover, some actual papers deal with the lipidomics approaches implemented into the monitoring of colistin therapy of tuberculosis (Anh, N.K.; Phat, N.K.; Yen, N.T.H.; Jayanti, R.P.; Thu, V.T.A.; Park, Y.J.; Cho, Y.-S.; Shin, J.-G.; Kim, D.-H.; Oh, J.-Y.; Long, N.-P. Comprehensive lipid profiles investigation reveals host metabolic and immune alterations during anti-tuberculosis treatment: Implications for therapeutic monitoring. Biomed. Pharmacother. 2023, 158, 114187. doi: 10.1016/j.biopha.2022.114187) and have shown significant changes of various individual lipids during the therapy.

All of the aforementioned facts were added into the manuscript to make it clearer.

Changes that have been made:

  1. Introduction

… therapy [43-46]. Nowadays, there is a significant rise in various system biology disciplines which enable to fulfill the aforementioned demands of clinicians. Metabolomics and lipidomics strategies represent the dominant ones, and are accompanied with the development of new drugs and exploring new potential biomarkers of various diseases [47-49]. Combination of these strategies with TDM (especially in case of ATB therapy) could be a very powerful and multivariate tool for unequivocal monitoring of the whole therapeutic protocol focused on the individual demands of the patient. Recently, a detail metabolomics study was performed on CST methanesulfonate treated Mycobacterium tuberculosis [50], which identified 22 significantly changed molecules. These findings may be helpful to clarify the mechanism of action of CST. Moreover, another very recent study by Carfrea et al. [51] demonstrated a significant impact of inhibition of fatty acid synthesis on overcoming CST resistance. This fact also confirms the increased interest of scientists in lipidomic approaches and their application in the clinical field [52]. Recently, the main interest is focused on lipidomics. In general, lipids are vital biomolecules deeply...

... diagnose human diseases [59]. Recently, lipid changes are often evaluated in septic patients [60-63]. It was demonstrated that sepsis is usually associated with the downregulation of lysophosphatidylcholine (LPC) levels, increased plasma free fatty acid (FA) and ceramide (Cer) species levels, or changes in the metabolism of polyunsaturated FAs (PUFA) in plasma [64].

Statement of the Reviewer 1:

The introduction could benefit from a clearer description of the current gaps in knowledge that this study aims to address. What specific questions about using TDM and lipidomics to manage antibiotic therapy in critically ill patients does this work attempt to answer?

Reply to the comments of the Reviewer 1:

Thank you for your comment and recommendation. The demanded information was added into the manuscript according to the recommendation of the reviewer. 

Changes that have been made:

  1. Introduction

... development of new drugs and exploring new potential biomarkers of various diseases [47-49]. Combination of these strategies with TDM (especially in case of ATB therapy) could be a very powerful and multivariate tool for unequivocal monitoring of the whole therapeutic protocol focused on the individual demands of the patient. Recently, a detail metabolomics study was performed on CST methanesulfonate treated Mycobacterium tuberculosis [50], which identified 22 significantly changed molecules. These findings may be helpful to clarify the mechanism of action of CST. Moreover, another very recent study by Carfrea et al. [51] demonstrated a significant impact of inhibition of fatty acid synthesis on overcoming CST resistance. This fact also confirms the increased interest of scientists in lipidomic approaches and their application in the clinical field [52]. Recently, the main interest is focused on lipidomics. In general, lipids are vital biomolecules deeply involved in diverse...

... Moreover, lipid profile helps to monitor the administration of clinical treatment and diagnose human diseases [59]. Recently, lipid changes are often evaluated in septic patients [60-63]. It was demonstrated that sepsis is usually associated with the downregulation of lysophosphatidylcholine (LPC) levels, increased plasma free fatty acid (FA) and ceramide (Cer) species levels, or changes in the metabolism of polyunsaturated FAs (PUFA) in plasma [64].

Statement of the Reviewer 1:

The lipidomic analysis methods are described in detail, but there is limited information provided about the validation and quality control measures used to ensure data reliability, such as the use of internal standards, calibration curves, and replicate analyses.

Reply to the comments of the Reviewer 1:

Thank you. As mentioned in the section "3.2.1 Chemicals and samples" and also in "3.2.3 Preparation of quality control (QC) samples and clinical samples", the commonly used internal standard mixture for lipidomic studies was used (SPLASH® LIPIDOMIX® Mass Spec Standard).

Calibration curves were not necessary because lipidomic methodology was only semi-quantitative and only peak areas obtained from analyses were used for statistical evaluation of individual lipid differences.

Changes that have been made:

..., adjusting the expected retention times (tR) of all lipids was necessary to set the optimal retention time window. To verify the quality of the analysis, pooled QC samples were analyzed after every three injections of samples. A characteristic LC-MS/MS TIC chromatogram of the separation of all lipids ...

Statement of the Reviewer 1:

The data processing and statistical analysis methods for the lipidomic results are not fully explained. More details on how raw data were handled, normalized, and analyzed would improve transparency and reproducibility.

Reply to the comments of the Reviewer 1:

Thank you for your comment. The whole procedure of the statistical analyses and data visualization was performed according to the procedure described in our previous paper published in 2024. To be more transparent and do not repeat the same information, the text in the section 3.2.4. Data and statistical analysis was modified appropriately and the relevant reference of our previous paper (Olesova et al., 2024) was added into the revised version of the manuscript.

Changes that have been made:

  1. Materials and Methods

3.2. Lipidomics

3.2.4. Data and statistical analysis

The data were acquired with the use of Analyst Software, and MultiQuant 2.1 (AB Sciex) software was used for the identification of lipids. All statistical analyses as well as data visualization were performed using R software (version 3.5.0) and the Metabol function package [81].

Statement of the Reviewer 1:

The generalizability and clinical relevance of the findings are not discussed. As this is a single-patient case study, it is important to consider the limitations of extrapolating the results to other patients and settings, and to frame the conclusions accordingly.

Reply to the comments of the Reviewer 1:

Thank you for your comment. However, the limitation and drawbacks of the presented study are summarized and discussed in section 2.4. Drawbacks of the case study and future perspectives. To make the text clearer, some parts of the text were modified or added into the manuscript.

Changes that have been made:

  1. Results and discussion

2.5. Drawbacks of the Case Study and Future Perspectives

The presented study highlighted the strong analytical potential of the developed methods for TDM of CST and lipidomics in critically ill patients. The main drawback of the lipidomic analysis is that only one critically ill patient was involved in the study. The single patient design of the study was not confronted and compared with untreated control samples. Therefore, the data cannot be generalized for all patients suffering from microbial infections. To improve the clinical output of the study, it will be necessary to monitor a large group of patients on CST therapy. Moreover, accurate stratification of patients into specific subgroups (e.g., responders vs. non-responders to the ATB treatment) will be of great importance. Such study design combining TDM and lipidomics will be able to identify and confirm specific individual lipid molecules serving as markers for successful ATB treatment of critically ill patients. We expect that in the near future, the incorporation of lipidomic approaches into the therapeutic protocols of bacterial infections, inflammation, and sepsis together with TDM of some last-resort ATBs (including CST) used in the therapy of these diseases will lead to improvements towards personalized medicine. Moreover, such multivariate data obtained during therapy will be associated with the possibility of an early intervention into therapy and offer better prediction of the cure prognosis. Similarly, we expect that the combination of TDM with lipidomics will significantly influence antibiotic stewardship, and so the responsible use of ATBs, with the goal of preserving their future effectiveness and safeguarding public health.

Statement of the Reviewer 1:

The potential impact of the findings on clinical practice is not addressed. Discussing how the TDM and lipidomic approaches could be implemented in real-world settings and what barriers need to be overcome would enhance the translational relevance of the conclusions.

Reply to the comments of the Reviewer 1:

Thank you for your valuable comment. The potential impact of the connection between TDM and lipidomics results was added and discussed in the manuscript according to the recommendation of the reviewer. We expect, that the added information will improve the quality of the manuscript.

Changes that have been made:

  1. Results and discussion

2.5. Drawbacks of the Case Study and Future Perspectives

The presented study highlighted the strong analytical potential of the developed methods for TDM of CST and lipidomics in critically ill patients. The main drawback of the lipidomic analysis is that only one critically ill patient was involved in the study. The single patient design of the study was not confronted and compared with untreated control samples. Therefore, the data cannot be generalized for all patients suffering from microbial infections. To improve the clinical output of the study, it will be necessary to monitor a large group of patients on CST therapy. Moreover, accurate stratification of patients into specific subgroups (e.g., responders vs. non-responders to the ATB treatment) will be of great importance. Such study design combining TDM and lipidomics will be able to identify and confirm specific individual lipid molecules serving as markers for successful ATB treatment of critically ill patients. We expect that in the near future, the incorporation of lipidomic approaches into the therapeutic protocols of bacterial infections, inflammation, and sepsis together with TDM of some last-resort ATBs (including CST) used in the therapy of these diseases will lead to improvements towards personalized medicine. Moreover, such multivariate data obtained during therapy will be associated with the possibility of an early intervention into therapy and offer better prediction of the cure prognosis. Similarly, we expect that the combination of TDM with lipidomics will significantly influence antibiotic stewardship, and so the responsible use of ATBs, with the goal of preserving their future effectiveness and safeguarding public health.

Statement of the Reviewer 1:

The results and discussion do not draw clear connections between the TDM and lipidomic findings. Integrating these two aspects more directly could yield insights into how colistin levels might influence lipid profiles and vice versa.

Reply to the comments of the Reviewer 1:

Thank you for your comment. We agree with the reviewer, that a clear connection between TDM and lipidomics findings are not clearly presented. According to the design of the study, it is difficult to give a clear answer, discuss the findings and generalize them. However, the discussion about TDM of colistin and obtained lipidomics data was added into the manuscript. 

Changes that have been made:

  1. Results and discussion

2.4. TDM of CST in View of Lipidomics Data

In this work, the combination of TDM with complex lipidomic profiling before, during, and after the administration of CST was realized for the first time. As mentioned in section 2.3 focused on results of lipidomics experiments, a significant drop in concentrations of all lipid classes at day 3 after CST administration was observed. This observation could probably be associated with CST administration. It is well known, that the steady state concentration of CST is typically obtained after three days [80], and reaching this pharmacokinetic parameter could be accompanied with changes in the concentration of some substances in biological fluids. However, this suggestion could not be confirmed in our study due to the limitations, such as the single-patient design of the study and the absence of untreated control samples. Further changes in the lipidomic profile and concentration levels of the investigated samples could also be associated with the concomitant ATB modalities used in the therapeutic protocol. To confirm and clearly evaluate potential lipidomic markers of CST therapy efficacy, it would be necessary to carry out a properly designed study including a higher number of probands and untreated control subjects. Recent results are able to present only a limited view on the association of CST plasma levels with changes in the lipid profile.

Statement of the Reviewer 1:

Comparisons to previous lipidomic studies in sepsis are made, but the discussion could benefit from more explicit consideration of how the current findings align with or diverge from prior work, and what new knowledge this study contributes.          

Reply to the comments of the Reviewer 1:

Thank you for your comment.  Our explorative study aims to investigate the lipid profile in the critically ill patient that underwent the colistin treatment. In the manuscript, we compared the results of our study with those of previous studies (e.g., lines 365-370, 383-384). However, it is challenging to compare our results with the other mentioned papers, as they reported significant changes in various lipid molecules that were not included in our analysis. We agree that more patient studies are needed to confirm our results.

Changes that have been made:

Statement of the Reviewer 1:

Potential limitations of the study, such as the single-patient design and lack of untreated control samples, are not addressed. Acknowledging and discussing these constraints would provide important context for interpreting the results.

Reply to the comments of the Reviewer 1:

Thank you for your comment. However, the limitation and drawbacks of the presented study are summarized and discussed in section 2.4. Drawbacks of the case study and future perspectives. To make the text clearer, some parts of the text were modified or added into the manuscript.

Changes that have been made:

  1. Results and discussion

2.5. Drawbacks of the Case Study and Future Perspectives

The presented study highlighted the strong analytical potential of the developed methods for TDM of CST and lipidomics in critically ill patients. The main drawback of the lipidomic analysis is that only one critically ill patient was involved in the study. The single patient design of the study was not confronted and compared with untreated control samples. Therefore, the data cannot be generalized for all patients suffering from microbial infections. To improve the clinical output of the study, it will be necessary to monitor a large group of patients on CST therapy. Moreover, accurate stratification of patients into specific subgroups (e.g., responders vs. non-responders to the ATB treatment) will be of great importance. Such study design combining TDM and lipidomics will be able to identify and confirm specific individual lipid molecules serving as markers for successful ATB treatment of critically ill patients.

Reviewer 2 Report

Comments and Suggestions for Authors

Introduction page 2, “Typically, it is performed only if the patient can benefit from it.” The authors need to correct this. TDM is mostly done for narrow therapeutic index drugs which includes antibiotics as well. It is not based on patient benefiting.

Overall the introduction is not well organized and too long. It is not a well-structured manuscript.

Table 1: It is not clear whether it is creatinine clearance or serum creatinine?

The description and presentation of the chemical structures in result section is inappropriate.

The figure-2 is showing a variable range above 1 µg/ml (MIC) to 2.5 µg/ml. What is the maximum safe concentration? Or minimum toxic concentration? Because in high doses colistin can cause nephrotoxicity or any other possible side effects. The therapeutic range of colistin is different in other reported studies as compared to current manuscript.

Figure 3: the peaks labelled as C. The peaks retention is shortened by 1 minute. In A and B, the peaks appear after 13 minutes while in C it is just after 12 min. There seems to be some problem in the method. It should not vary by that much time. 

In result section “Elevated levels of ALT and AST are often indicators of liver damage or disease, which is common when multiple antibiotics are used” give reference of the drugs used regarding liver damage (toxicity). Generalized statements are discouraged in scientific manuscripts.

Table 2 is showing the gradient elution; 15.50 to 15.60 min is very short time for gradient elution.

Comments on the Quality of English Language

The language is not appropriate for a scientific manuscript. 

Author Response

Dear Editors,

Dear Referees,

At the beginning, I would like to thank you for your interest in our manuscript (manuscript ID: pharmaceuticals-3027108), “Implementation of modern therapeutic drug monitoring and lipidomics approaches in clinical practice: A case study with colistin treatment” (authors Gerhardtova,I., Cizmarova,I., Jankech,T., Olesova,D., Jampilek,J., Parrak,V., Nemergutova,K., Sopko,L., Piestansky,J.*, Kovac,A.*), which has been submitted to your Journal (Pharmaceuticals) – special issue: "Innovative Tools for Drug Analysis and Therapeutic Drug Monitoring (TDM)".

On 17 May 2024 I received the reports to the manuscript. Please find my point-by-point reply to the comments and to the proposed amendments attached to this message. The paper was modified following the recommendations of the referees. The changes that have been made in the text are clearly marked. The "Response to Reviews" is included to this submission.

Finally, I would like to express my thanks to you and to the referees for the kind handling, valuable advice and suggestions on the manuscript. I look forward to further cooperation.

Sincerely Yours,

Juraj Piestansky & Andrej Kovac

(Corresponding author)

Bratislava, May 30, 2024

Assoc. prof. Juraj Piestansky

Department of Galenic Pharmacy,

Faculty of Pharmacy,

Comenius University,

Odbojarov 10,

SK-832 32 Bratislava,

Slovak Republic

Phone: **421-2-50 117 250

E-mail: piestansky@fpharm.uniba.sk

Dr. Andrej Kovac

Institute of Neuroimmunology,

Slovak Academy of Sciences,

Dubravska cesta 9,

845 10 Bratislava,

Slovak Republic

E-mail: andrej.kovac@savba.sk

Reviewer #2:

Statement of the Reviewer 2:

Introduction page 2, “Typically, it is performed only if the patient can benefit from it.” The authors need to correct this. TDM is mostly done for narrow therapeutic index drugs which includes antibiotics as well. It is not based on patient benefiting.

Reply to the comments of the Reviewer 2:

Thank you for your valuable comment. We agree with the reviewer. It is true, that the therapeutic drug monitoring is performed in case of drugs with narrow therapeutic index. The information presented in section 1. Introduction was modified according to the recommendation of the reviewer. The confusing information dealing with the patient benefiting was deleted and the text was shortened.

Changes that have been made:

  1. Introduction

In general, TDM represents a measurement of drug concentration in the body fluids or tissues, which is performed to optimize the patient's therapy outcome [13]. Typically, it is performed for drugs with narrow therapeutic index, drugs with a well-defined relationship between concentration and effect, and drugs with large inter- or intra-individual distribution or clearance differences. A drug is a potential candidate for TDM, if there is a large interindividual pharmacokinetic variability, and if the therapeutic effect of the drug cannot be adequately and easily measured [14]. CST meets the mentioned criteria, and measurement of its concentration in critically ill patients is highly demanded by the clinicians to control and optimize the drug dosing regimen.

Statement of the Reviewer 2:

Overall the introduction is not well organized and too long. It is not a well-structured manuscript.

Reply to the comments of the Reviewer 2:

Thank you for your opinion. The section 1. Introduction was reorganized to make the manuscript clearer. Some important facts were added to the manuscript with appropriate citations of relevant scientific papers. The cited papers were added into the section References.

Changes that have been made:

  1. Introduction

Antimicrobial resistance represents a serious threat to human health around the whole world in the 21st century [1]. Particular difficulties are associated with antibiotic resistant Gram-negative pathogens, such as Escherichia coli, Klebsiella pneumoniae, Acinetobacter baumannii, and Pseudomonas aeruginosa [1-4]. The rise of antibiotic resistance led to repurposing of some old antibiotics (ATBs) such as polymyxins (including colistin), which are recently used as the last therapeutic option for infections caused by the aforementioned Gram-negative bacteria [5, 6]. Colistin (CST, or polymyxin E) is a cyclic lipopeptide with a narrow antibacterial spectrum. Generally, it has two forms, i.e., colistin A (CST A, or polymyxin E1) and colistin B (CST B, or polymyxin E2). Differences in the chemical structures of these two forms are shown in Figure S1, Supplementary material. Although polymyxins entered the clinical praxis in the late 1950s, their mechanism of action is still not well understood [7, 8]. It is expected that polymyxins target lipid A, a specific component of the lipopolysaccharide present on the bacterial outer membrane [9]. Moreover, information on physicochemical and pharmacological properties of these ATBs is limited, which significantly affects their appropriate and safe use in clinical practice, especially in critically ill patients [10]. In such cases, therapeutic drug monitoring (TDM) of ATBs seems to be a promising tool to set optimal therapy management of critically ill patients, which includes optimization of dosage regimens, minimization of unwanted side effects, and prevention of bacterial resistance. Here, it is necessary to mention that achieving the therapeutic effect of ATBs is challenging, e.g., in morbidly obese patients, in patients with catheters, or in increased renal and/or hepatic function [11]. The correct dosage of ATBs is imperative to ensure their adequate exposure [12].

In general, TDM represents a measurement of drug concentration in the body fluids or tissues, which is performed to optimize the patient's therapy outcome [13]. Typically, it is performed for drugs with narrow therapeutic index, drugs with a well-defined relationship between concentration and effect, and drugs with large inter- or intra-individual distribution or clearance differences. A drug is a potential candidate for TDM, if there is a large interindividual pharmacokinetic variability, and if the therapeutic effect of the drug cannot be adequately and easily measured [14]. CST meets the mentioned criteria, and measurement of its concentration in critically ill patients is highly demanded by the clinicians to control and optimize the drug dosing regimen.

The TDM procedure is composed of three phases: i) pre-analytical phase, ii) analytical phase, and iii) post-analytical phase (proper clinical interpretation of the measured data). The pre-analytical phase is accompanied with appropriate planning of the sampling, sample collection, and storing. Here, the knowledge about the exact time of drug administration is critical for reliable interpretation of measured data. In general, samples are collected at steady state [15]. The results from the laboratory should be available within a short period of time, in the optimal situation before the administration of the next dose [16]. In case of uncertainty about the therapeutic efficacy of the drug, samples are collected just before the administration of the next dose [17].

The analytical phase usually consists in the implementation of chromatographic (especially high-performance liquid chromatography, HPLC) and immunologic methods. However, they have some limitations including the lack of standardization of work procedures, long analysis time, high costs, and demands on complex preparation and pre-treatment of samples. The above-mentioned weakness of the convenient methods could be overcome by new developing technologies based on capillary zone electrophoresis (CZE) [18] or biosensors [19]. However, the implementation of such analytical approaches in the clinical environment is not yet fully established. CZE is a promising alternative method for the analysis of drugs and their metabolites in biological fluids. It offers several advantages in TDM, i.e., relatively fast analysis time, simple instrumentation, environmental friendliness, better resolution, high separation efficiency allowing multicomponent analysis, and low cost of analysis compared to HPLC (only a small amount of solvent is required, and capillaries are relatively inexpensive). The drawback of CZE is the lack of sensitivity and suboptimal detection limits. However, these problems can be solved by appropriate sample pretreatment and/or combination with selective and sensitive detection (e.g., mass spectrometry, MS). CZE methods are usually used only for newer drugs in TDM, as HPLC assays or immunoassays are available for older drugs [20]. The dominant position of HPLC methods in TDM was also observed in case of CST determination, where the instrumental approaches based on hyphenation of liquid chromatography (LC) with fluorescence (FLD) [21–24] and/or MS detection [25–41] are the most common. There is only one CZE-MS/MS method for TDM of CST described in the scientific literature, and this approach was developed by our research group [42].

Recent trends in clinical practice are oriented toward personalized medicine. Such an approach typically demands a fundamental understanding of the disease, identification of drug targets for therapy, and the discovery of relevant biomarkers of the disease or monitoring the effectiveness of drug treatment leading to clinical follow-up in medical therapy [43-46]. Nowadays, there is a significant rise in various fields of systems biology, which enable to fulfill the aforementioned demands of clinicians. Metabolomic and lipidomic strategies are dominant ones and are accompanied by the development of new drugs and exploring new potential biomarkers of various diseases [47-49]. Combination of these strategies with TDM (especially in case of ATB therapy) could be a very powerful and multivariate tool for unequivocal monitoring of the whole therapeutic protocol focused on the individual demands of the patient. Recently, a detail metabolomics study was performed on CST methanesulfonate treated Mycobacterium tuberculosis [50], which identified 22 significantly changed molecules. These findings may be helpful to clarify the mechanism of action of CST. Moreover, another very recent study by Carfrea et al. [51] demonstrated a significant impact of inhibition of fatty acid synthesis on overcoming CST resistance. This fact also confirms the increased interest of scientists in lipidomic approaches and their application in the clinical field [52]. Recently, the main interest is focused on lipidomics. In general, lipids…

References

  1. Antimicrobial Resistance Collaborators. Global burden of bacterial antimicrobial resistance in 2019: a systematic analysis. Lancet. 2022, 399, P629-655. doi: 10.1016/S0140-6736(21)02724-0.
  2. Feretzakis, G.; Loupelis, E.; Sakagianni, A.; Skarmoutsou, N.; Michelidou, S.; Velentza, A.; Martsoukou, M.; Valakis, K.; Petropoulou, S.; Koutalas, E. A 2-year single-centre audit on antibiotic resistance of Pseudomonas aeruginosa, Acinetobacter baumannii and Klebsiella pneumoniae strains from intensive care unit and other wards in a general public hospital in Greece. Antibiotics. 2019, 8, 62. doi: 10.3390/antibiotics8020062.
  3. Mancuso, G.; Midiri, A.; Gerace, E.; Biondo, C. Bacterial antibiotic resistance: The most critical pathogens. Pathogens. 2021, 10, 1310. doi: 10.3390/pathogens10101310.
  4. Kumar, N.R.; Balraj, T.A.; Kempegowda, S.N.; Prashant, A.; Multidrug-resistant sepsis: A critical healthcare challenge. Antibiotics. 2024, 13, 46. doi: 10.3390/antibiotics13010046
  5. Velkov, T.; Thompson, P.E.; Azad, M.A.K.; Roberts, K.D.; Bergen, P.J. History, Chemistry, and Antibacterial Spectrum. In Polymyxin Antibiotics: From Laboratory Bench to Bedside, 1th ed.; Li, J., Nation, R.L., Kaye, K.S., Eds.; Springer Nature: Cham, Switzerland, 2019; pp.15-36.
  6. Li, J.; Nation, R.L.; Turnidge, J.D.; Milne, R.W.; Coulthard, K.; Rayner, C.R.; Paterson, D.L. Colistin: the re-emerging antibiotic for multidrug-resistant Gram-negative bacterial infections. Lancet Infect Dis. 2006, 6, 589-601. doi: 10.1016/S1473-3099(06)70580-1.
  7. Poirel, L.; Jayol, A.; Nordmann, P. Polymyxins: Antibacterial activity, susceptibility testing, and resistance mechanisms encoded by plasmids or chromosomes. Clin. Microbiol. Rev. 2017, 30, 557-596. doi: 10.1128/CMR.00064-16.
  8. Andrade, F.F.; Silva, D.; Rodrigues, A.; Pina-Vaz, C. Colistin update on its mechanism of action and resistance, present and future challenges. Microorganisms, 2020, 8, 1716. doi: 10.3390/microorganisms8111716.
  9. Manioglu, S.; Modaresi, S.M.; Ritzmann, N.; Thoma, J.; Overall, S.A.; Harms, A.; Upert, G.; Luther, A.; Barnes, A.B.; Obrecht, D.; Müller, D.J.; Hiller, S. Antibiotic polymyxin arranges lipopolysaccharide into crystalline structures to solidify the bacterial membrane. Nat. Commun. 2022, 13, 6195. doi: 10.1038/s41467-022-33838-0.
  10. Rychlíčková, J.; Kubíčková, V.; Suk, P.; Urbánek, K. Challenges of colistin use in ICU and therapeutic drug monitoring: a literature review. Antibiotics. 2023, 12, 437. doi: 10.3390/antibiotics12030437.
  11. Meng, L.; Mui, E.; Ha, D.R.; Stave, C.; Deresinski, S.C.; Holubar, M. Comprehensive guidance for antibiotic dosing in obese adults: 2022 update. Pharmacotherapy, 2023, 43, 226-246. doi: 10.1002/phar.2769.
  12. Koch, B.C.P.; Muller, A.E.; Hunfeld, N.G.M.; de Winter, B.C.M.; Ewoldt, T.M.J.; Abdulla, A.; Endeman, H. Therapeutic drug monitoring of antibiotics in critically ill patients: Current practice and future perspectives with focus on clinical outcome. Ther. Drug Monit. 2022, 44, 11-18. doi: 10.1097/FTD.0000000000000942.
  13. Dasgupta, A. Introduction to therapeutic drug monitoring: Frequently and less frequently monitored drugs. In Therapeutic Drug Monitoring, 1st; Dasgupta, A., Eds., Academic Press: San Diego, CA, 2012; pp. 1–29.
  14. Roberts, J.A.; Ross, N.; Peterson, D.L..; Martin, J.H. Therapeutic drug monitoring of antimicrobials. Br. J. Clin. Pharmacol. 2012, 73, 27-36. doi: 10.1111/j.1365-2125.2011.04080.x.
  15. Mann, K.; Hiemke, C.; Schmidt, L.G.; Bates, D.W. Appropriateness of therapeutic drug monitoring for antidepressants in routine psychiatric inpatient care. Ther. Drug Monit. 2006, 28, 83-88. doi: 10.1097/01.ftd.0000189897.16307.65.
  16. Gross, A.S. Best practice in therapeutic drug monitoring. Br. J. Clin. Pharmacol. 2001, 52, 5S-10S. doi: 10.1046/j.1365-2125.2001.0520s1005.x.
  17. Hammet-Stabler, C.A., Dasgupta, A. Therapeutic drug monitoring data: A concise guide, 3rd ed.; AACC Press: Washington, DC, USA, 2007; 241 p.
  18. Thormann, W. Progress of capillary electrophoresis in therapeutic drug monitoring and clinical and forensic toxicology. Ther. Drug Monit. 2002, 24, 222-231. doi: 10.1097/00007691-200204000-00004.
  19. Ates, H.C.; Roberts, J.A.; Lipman, J.; Cass, A.E.G.; Urban, G.A.; Dincer, C. On-site therapeutic drug monitoring. Trends Biotechnol. 2020, 38, 1262-1277. doi: 10.1016/j.tibtech.2020.03.001.
  20. Shihabi, Z.K. Capillary electrophoresis for the determination of drugs in biological fluids. In Handbook of Analytical Separations, 1th ed.; Hempel, G., Ed.; Elsevier: Amsterdam, Netherlands, 2004; Volume 5, pp.77-94.
  21. Le Brun, P.P.H.; de Graaf, A.I.; Vinks, A.A.T.M.M. High-performance liquid chromatographic method for the determination of colistin in serum. Drug Monit. 2000, 22, 589–593. doi: 10.1097/00007691-200010000-00014.
  22. Li, J.; Milne, R.W.; Nation, R.L.; Turnidge, J.D.; Coulthard, K.; Johnson, D.W. A simple method for the assay of colistin in human plasma, using pre-column derivatization with 9-fluorenylmethyl chloroformate in solid-phase extraction cartridges and reversed-phase high-performance liquid chromatography. J. Chromatogr. B. 2001, 761, 167–175. doi: 10.1016/S0378-4347(01)00326-7.
  23. Reed, M.D.; Stern, R.C.; O’Riordan, M.A.; Blumer, J.L. The pharmacokinetics of colistin in patients with cystic fibrosis. J. Clin. Pharmacol. 2001, 41, 645–654. doi:10.1177/00912700122010537.
  24. Chepyala, D.; Tsai, I.-L.; Sun, H.-Y.; Lin, S.-W.; Kuo, C.-H. Development and validation of a high-performance liquid chromatography-fluorescence detection method for the accurate quantification of colistin in human plasma. J. Chromatogr. B. 2015, 980, 48–54. doi: 10.1016/j.jchromb.2014.12.015.
  25. Jansson, B.; Karvanen, M.; Cars, O.; Plachouras, D.; Friberg, L.E. Quantitative analysis of colistin A and colistin B in plasma and culture medium using a simple precipitation step followed by LC/MS/MS. J. Pharm. Biomed. Anal. 2009, 49, 760–767. doi: 10.1016/j.jpba.2008.12.016.
  26. Ma, Z.; Wang, J.; Gerber, J.P.; Milne, R.W. Determination of colistin in human plasma, urine and other biological samples using LC–MS/MS. J. Chromatogr. B. 2008, 862, 205–212. doi: 10.1016/j.jchromb.2007.12.009.
  27. Gobin, P.; Lemaître, F.; Marchand, S.; Couet, W.; Olivier, J.-C. Assay of colistin and colistin methanesulfonate in plasma and urine by liquid chromatography-tandem mass spectrometry. Antimicrob. Agents Chemother. 2010, 54, 1941–1948. doi: 10.1128/AAC.01367-09.
  28. Dotsikas, Y.; Markopoulou, C.K.; Koundourellis, J.E.; Loukas, Y.L. Validation of a novel LC-MS/MS method for the quantitation of colistin A and B in human plasma. J. Sep. Sci. 2011, 34, 37–45. doi: 10.1002/jssc.201000680.
  29. Gikas, E.; Bazoti, F.N.; Katsimardou, M.; Anagnostopoulos, D.; Papanikolaou, K.; Inglezos, I.; Skoutelis, A.; Daikos, G.L.; Tsarbopoulos, A. Determination of colistin A and colistin B in human plasma by UPLC–ESI high resolution tandem MS: Application to a pharmacokinetic study. J. Pharm. Biomed. Anal. 2013, 83, 228–236. doi: 10.1016/j.jpba.2013.05.008.
  30. Tsai, I.-L.; Sun, H.-Y.; Chen, G.-Y.; Lin, S.-W.; Kuo, C.-H. Simultaneous quantification of antimicrobial agents for multidrug-resistant bacterial infections in human plasma by ultra-high-pressure liquid chromatography–tandem mass spectrometry. Talanta. 2013, 116, 593–603. doi: 10.1016/j.talanta.2013.07.043.
  31. Leporati, M.; Bua, R.O.; Mariano, F.; Carignano, P.; Stella, M.; Biancone, L.; Vincenti, M. Determination by LC–MS/MS of colistins A and B in plasma and ultrafiltrate from critically ill patients undergoing continuous venovenous hemodiafiltration. Ther. Drug Monit. 2014, 36, 182–191. doi: 10.1097/FTD.0b013e3182a8997c.
  32. Mercier, T.; Tissot, F.; Gardiol, C.; Corti, N.; Wehrli, S.; Guidi, M.; Csajka, C.; Buclin, T.; Couet, W.; Marchetti, O.; Decosterd, L.A. High-throughput hydrophilic interaction chromatography coupled to tandem mass spectrometry for the optimized quantification of the anti-Gram-negatives antibiotic colistin A/B and its pro-drug colistimethate. J. Chromatogr. A. 2014, 1369, 52–63. doi: 10.1016/j.chroma.2014.09.063.
  33. Zhao, M.; Wu, X.-J.; Fan, Y.-X.; Guo, B.-N.; Zhang, J. Development and validation of a UHPLC–MS/MS assay for colistin methanesulphonate (CMS) and colistin in human plasma and urine using weak-cation exchange solid-phase extraction. J. Pharm. Biomed. Anal. 2016, 124, 303–308. doi: 10.1016/j.jpba.2016.02.045.
  34. Bihan, K.; Lu, Q.; Enjalbert, M.; Apparuit, M.; Langeron, O.; Rouby, J.-J.; Funck-Brentano, C.; Zahr, N. Determination of colistin and colistimethate levels in human plasma and urine by high-performance liquid chromatography–tandem mass spectrometry. Ther. Drug Monit. 2016, 38, 796–803. doi: 10.1097/FTD.0000000000000345.
  35. Cangemi, G.; Barco, S.; Castagnola, E.; Tripodi, G.; Favata, F.; D’Avolio, A. Development and validation of UHPLC–MS/MS methods for the quantification of colistin in plasma and dried plasma spots. J. Pharm. Biomed. Anal. 2016, 129, 551–557. doi: 10.1016/j.jpba.2016.08.002.
  36. Binhashim, N.H.; Alvi, S.N.; Hammami, M.M. LC-MS/MS method for determination of colistin in human plasma: Validation and stability studies. IJAMSC. 2021, 09, 1–11. doi: 10.4236/ijamsc.2021.91001.
  37. Matar, K.M.; Al-Refai, B. Quantification of colistin in plasma by liquid chromatography-tandem mass spectrometry: Application to a pharmacokinetic study. Sci Rep. 2020, 10, 8198. doi: 10.1038/s41598-020-65041-w.
  38. Qi, B.; Gijsen, M.; Van Brantegem, P.; De Vocht, T.; Deferm, N.; Abza, G.B.; Nauwelaerts, N.; Wauters, J.; Spriet, I.; Annaert, P. Quantitative determination of colistin A/B and colistin methanesulfonate in biological samples using hydrophilic interaction chromatography tandem mass spectrometry. Drug Test. Anal. 2020, 12, 1183–1195. doi: 10.1002/dta.2812.
  39. Yuan, H.; Yu, S.; Chai, G.; Liu, J.; Zhou, Q.T. An LC-MS/MS method for simultaneous analysis of the cystic fibrosis therapeutic drugs colistin, ivacaftor and ciprofloxacin. J. Pharm. Anal. 2021, 11, 732-738. doi: 10.1016/j.jpha.2021.02.004.
  40. Wang, X.; Sun, Q.; Li, X.; Wang, G.; Xing, B.; Li, Z. Novel method for determination of colistin sulfate in human plasma by high-performance liquid chromatography-tandem mass spectrometry and its clinical applications in critically ill patients. J. Pharmacol. Toxicol. Methods. 2024, 127, 107502. doi: 10.1016/j.vascn.2024.107502.
  41. Neef, S.K.; Hinderer, A.D.; Arbash, W.; Kinzig, M.; Sörgel, F.; Wunder, C.; Schwab, M.; Hofmann, U. A high performance liquid chromatography-tandem mass spectrometry assay for therapeutic drug monitoring of 10 drug compounds commonly used for antimicrobial therapy in plasma and serum of critically ill patients: Method optimization, validation, cross-validation and clinical application. Clin. Chim. Acta. 2024, 559, 119690. doi: 10.1016/j.cca.2024.119690
  42. Cizmarova, I.; Parrak, V.; Secnik jr., P.; Secnik, P.; Sopko, L.; Nemergutova, K.; Kovac, A.; Mikus, P.; Piestansky, J. A simple and green capillary electrophoresis-mass spectrometry method for therapeutic drug monitoring of colistin in clinical plasma samples. Helyion. 2023, 9, e23111. doi: 10.1016/j.heliyon.2023.e23111.
  43. Lee, S.; Mun, S.; Lee, J.; Kang, H.-G. Discovery and validation of protein biomarkers for monitoring the effectiveness of drug treatment for major depressive disorder. J. Psychiatr. Res. 2024, 169, 7-13. doi: 10.1016/j.jpsychires.2023.11.005.
  44. Bodaghi, A.; Fattahi, N.; Ramazani, A. Biomarkers: Promising and valuable tools towards diagnosis, prognosis and treatment of Covid-19 and other diseases. Helyion. 2023, 9, e13323. doi: https://doi.org/10.1016/j.heliyon.2023.e13323.
  45. Aronson, J.K.; Ferner, R.E. Biomarkers – a general review. Curr. Protoc. Pharmacol. 2017, 9.23.1-9.23.17. doi: 10.1002/cpph.19.
  46. Jafarzadeh, L.; Khakpoor-Koosheh, M.; Mirzaei, H.; Mirzaei, H.R. Biomarkers for predicting the outcome of various cancer immunotherapies. Crit. Rev. Oncol. Hematol. 2021, 157, 103161. doi: 10.1016/j.critrevonc.2020.103161.
  47. Astarita, G.; Kelly, R.S.; Lasky-Su, J. Metabolomics and lipidomics strategies in modern drug discovery and development. Drug Discov. Today. 2023, 28, 103751. doi: 10.1016/j.drudis.2023.103751.
  48. Shi, W.; Cheng, Y.; Zhu, H.; Zha, L. Metabolomics and lipidomics in non-small cell lung cancer. Clin. Chim. Acta. 2024, 555, 117823. doi: 10.1016/j.cca.2024.117823.
  49. Liu, X.; Zhang, M.; Cheng, X.; Liu, X.; Sun, H.; Guo, Z; Li, J.; Tang, X.; Wang, Z.; Sun, W.; Zhang, Y.; Ji, Z. LC-MS-Based Plasma Metabolomics and Lipidomics Analyses for Differential Diagnosis of Bladder Cancer and Renal Cell Carcinoma. Front. Oncol. 2020, 10, 717. doi: 10.3389/fonc.2020.00717.
  50. Koen, N.; van Breda, S.V.; Loots, D.T. Metabolomics of colistin methanesulfonate treated Mycobacterium tuberculosis. Tuberculosis. 2018, 111, 154-160. doi: 10.1016/j.tube.2018.06.008.
  51. Carfrae, L.A.; Rachwalski, K.; French, S.; Gorzevich, R.; Seidel, L.; Tsai, C.N.; Tu, M.M.; MacNair, C.R.; Ovchinnikova, O.G., Clarke, B.R.; Whitfield, C.; Brown, E.D. Inhibiting fatty acid synthesis overcomes colistin resistance. Nat. Microbiol. 2023, 8, 1026-1038. doi: 10.1038/s41564-023-01369-z.
  52. Meikle, T.G.; Huynh, K.; Giles, C.; Meikle, P.J. Clinical lipidomics: realizing the potential of lipid profiling. J. Lipid. Res. 2021, 62, 100127. doi: 10.1016/j.jlr.2021.100127.
  53. Züllig, T.; Trötzmüller, M.; Köfeler, H.C. Lipidomics from Sample Preparation to Data Analysis: A Primer. Anal. Bioanal. Chem. 2020, 412, 2191– doi: 10.1007/s00216-019-02241-y.

….

Statement of the Reviewer 2:

Table 1: It is not clear whether it is creatinine clearance or serum creatinine?

Reply to the comments of the Reviewer 2:

Thank you for your suggestion. Creatinine in the Table 1 represents serum creatinine. This information was added into the Table 1 to make the parameter clearer.

Changes that have been made:

  1. Results and discussion

2.1. Case presentation

Table 1. Overview of routinely determined biochemical parameters during the hospitalization and follow up of the patient.

Clinical parameter

Day 1

Day 2*1

Day 3

Day 4

Day 5

Day 6

Day 7

Day 8*2

Day 9

Day 10

Day 11

Day 12

Follow up *3

Serum creatinine (μmol/L)

70.6

89.9

89.7

83.9

109.7

129

125

111

84

85.7

92.3

89.5

76.2

GF (mL/s)

1.89

1.56

1.56

1.69

1.22

1.00

1.04

1.20

1.69

1.65

1.51

1.56

1.83

ALT (μkat/L)

1.27

n.d.

3.46

2.00

5.00

3.24

2.66

2.37

n.d.

n.d.

1.12

0.53

0.35

AST (μkat/L)

1.46

n.d.

5.27

1.87

8.34

3.40

2.21

1.83

n.d.

n.d.

0.78

0.57

0.43

GGT (μkat/L)

3.79

n.d.

7.10

6.31

9.11

8.66

8.04

7.61

n.d.

n.d.

4.47

2.78

0.77

ALP (μkat/L)

2.28

n.d.

5.14

4.37

6.32

5.60

5.55

5.30

n.d.

n.d.

3.66

3.56

1.89

Albumin (g/L)

26.6

28.6

27.0

29.5

29.6

33.0

33.0

35.9

n.d.

n.d.

33.0

38.0

40.7

CRP (mg/L)

335

442

364.8

338

313.4

263.5

201.0

153.4

80.2

49.5

37.26

8.35

5.37

Glucose (mmol/L)

n.d.

5.88

5.77

16.19

16.4

4.9

5.18

6.11

n.d.

n.d.

5.32

5.82

5.58

Bilirubin (mmol/L)

8.4

n.d.

11.8

8.6

8.8

6.9

7.4

6.5

n.d.

n.d.

6.1

n.d.

n.d.

Na (mmol/L)

134

145

142

130

131

138

136

133

135

137

135

143

139

K (mmol/L)

3.41

3.94

3.80

3.92

4.38

4.60

4.40

4.10

4.06

3.80

3.88

3.40

3.48

Hematocrit (%)

26.4

24.4

25.1

27.3

29.6

30.7

32.3

34

32.8

32.4

30

30

25.3

Abbreviations: GF (glomerular filtration); ALT (alanine transaminase); AST( aspartate transaminase); GGT (gamma-glutamyltransferase); ALP (alkaline phosphatase); CRP (C-reactive protein); Na (sodium); K (potassium), n.d.( not determined)

*1 The day when colistin was first administered.

*2 The day when colistin was last administered.

*3 Biochemical parameters investigated after the follow-up examination of the patient on outpatient basis 51 days after hospitalization.

Statement of the Reviewer 2:

The description and presentation of the chemical structures in result section is inappropriate.

Reply to the comments of the Reviewer 2:

We agree with the reviewer. Original Figure 1 was deleted from the main part of the manuscript and was moved to the Supplementary material. Figure S-1 was added to the Supplementary material. The number of all other Figures in the main manuscript and also in the Supplementary material was renumbered. Moreover, description of the structures was moved to the section 1. Introduction.

Changes that have been made:

  1. Introduction

… the last therapeutic option for infections caused by the aforementioned Gram-negative bacteria [5, 6]. Colistin (CST, or polymyxin E) is a cyclic lipopeptide with a narrow antibacterial spectrum. Generally, it has two forms, i.e., colistin A (CST A, or polymyxin E1) and colistin B (CST B, or polymyxin E2). Differences in the chemical structures of these two forms are shown in Figure S1, Supplementary material. Although polymyxins entered the clinical praxis in the late 1950s, their mechanism of action…

  1. Results and discussion

2.2. Therapeutic Drug Monitoring of CST

              From the clinical point of view, CST has an extremely narrow therapeutic index, and the plasma concentration required for antibacterial activity can be similar to that which predisposes it to nephrotoxicity [71, 72]. According to Regenthal et al. [73], the therapeutic plasma concentration range is 1–5 μg/mL. Therefore, the implementation of TDM is essential in such cases. As mentioned previously, blood samples were collected every morning before the intravenous administration of the drugs. This sampling procedure was in accordance with the TDM guidelines [74]. This sampling strategy was chosen due to the severe medical condition of the patient, but the optimal time of sampling was not respected.

Article

Implementation of modern therapeutic drug monitoring and lipidomics approaches in clinical practice: A case study with colistin treatment

Ivana Gerhardtova 1,2, Ivana Cizmarova 3,4, Timotej Jankech 1,2, Dominika Olesova 1,5, Josef Jampilek 1,2, Vojtech Parrak 1,6, Kristina Nemergutova 6, Ladislav Sopko 6, Juraj Piestansky 1,4,7,* and Andrej Kovac 1, 8*

1    Institute of Neuroimmunology, Slovak Academy of Sciences, Dubravska cesta 9, 845 10 Bratislava, Slovak Republic; timotej.jankech@gmail.com (T.J.); ivka.gerhardtova@gmail.com (I.G.); josef.jampilek@gmail.com (J.J.); vojtech.parrak@savba.sk (V.P.); andrej.kovac@savba.sk (A.K.)

2    Department of Analytical Chemistry, Faculty of Natural Sciences, Comenius University Bratislava, Ilkovicova 6, 842 15 Bratislava, Slovak Republic

3    Department of Pharmaceutical Analysis and Nuclear Pharmacy, Faculty of Pharmacy, Comenius University Bratislava, Odbojarov 10, 832 32 Bratislava, Slovak Republic; ivana.cizmarova@fpharm.uniba.sk (I.C.)

4    Toxicological and Antidoping Center, Faculty of Pharmacy, Comenius University Bratislava, Odbojarov 10, 832 32 Bratislava, Slovak Republic

5    Institute of Experimental Endocrinology, Biomedical Research Center SAS, Dubravska cesta 9, 845 10 Bratislava, Slovak Republic; dominika.olesova@savba.sk (D.O.)

6    Clinic of Hematology and Transfusiology, St Cyril and Methodius Hospital, Antolska 11, 851 07, Bratislava, Slovak Republic; sopko.ladislav@gmail.com (L.S.); kika.nemergutova@gmail.com (K.N.)

7    Department of Galenic Pharmacy, Faculty of Pharmacy, Comenius University Bratislava, Odbojarov 10, 832 32 Bratislava, Slovak Republic; piestansky@fpharm.uniba.sk (J.P.)

8  Department of Pharmacology and Toxicology, University of Veterinary Medicine and Pharmacy in Kosice, Komenského 68/73, 041 81, Kosice, Slovak Republic

*   Correspondence: piestansky@fpharm.uniba.sk (J.P.); andrej.kovac@savba.sk (A.K.)

(Supplementary material)

Table of content:

Figure S1: Chemical structure of colistin A (polymyxin E1) and colistin B (polymyxin E2).

Figure S2: A complex overview of concentration changes in selected individual lipids during the hospitalization of a critically ill patient and his treatment with multiple ATBs including colistin.

Table S1. Concentration levels of individual lipids determined during the colistin therapy.

Figure S1. Chemical structure of colistin A (polymyxin E1) and colistin B (polymyxin E2).

Statement of the Reviewer 2:

The figure-2 is showing a variable range above 1 µg/ml (MIC) to 2.5 µg/ml. What is the maximum safe concentration? Or minimum toxic concentration? Because in high doses colistin can cause nephrotoxicity or any other possible side effects. The therapeutic range of colistin is different in other reported studies as compared to current manuscript.

Reply to the comments of the Reviewer 2:

Li et al. (doi: 10.1016/S1473-3099(06)70580-1, reference 45 in the manuscript) stated that the minimal inhibition concentration should be 1 ug/mL). According to the Institute of Clinical and Laboratory Standards, it is necessary, that the plasma concentration of colistin achieve the values from 1 to 5 ug/mL (doi: 10.1023/A:1009935116877). It can be therefore stated, that the maximum safe concentration of colistin is 5 ug/mL. The determined concentration in plasma samples depends on the dosing regimen. In our case, the samples were collected in the morning right before the administration of the dose, therefore, the determined concentration in our case were lower than those reported in other studies. The sampling procedure is in detail described in the original manuscript. However, to make the manuscript clearer, the information dealing with the plasma colistin interval was added into the manuscript ad relevant references were added to the section References.

Changes that have been made:

  1. Results and discussion

2.2. Therapeutic Drug Monitoring of CST

              From the clinical point of view, CST has an extremely narrow therapeutic index, and the plasma concentration required for antibacterial activity can be similar to that which predisposes it to nephrotoxicity [71, 72]. According to Regenthal et al. [73], the therapeutic plasma concentration range is 1–5 μg/mL. Therefore, the implementation of TDM is essential...

Statement of the Reviewer 2:

Figure 3: the peaks labelled as C. The peaks retention is shortened by 1 minute. In A and B, the peaks appear after 13 minutes while in C it is just after 12 min. There seems to be some problem in the method. It should not vary by that much time. 

Reply to the comments of the Reviewer 2:

Thank you for your suggestion. The method used for the TDM procedure was based on CZE-MS/MS. CZE is characterized by lower robustness as the convenient LC methods. The result of lower robustness is the change of the migration time. However, as can be seen from the Figure 2 (Figure 3 in the original version of the manuscript), the fluctuation of the migration time affects all of the investigated analytes. Moreover, IS was used in the method. The use of IS minimizes the negative effects during the separation procedure, i.e., minimizes fluctuation of peak area and also reduces the negative impact of the robustness. In such cases the standardized migration time can be used which is characterized as a ratio of migration time of the investigated analyte (here CST A, or CST B) to the migration time of the IS (polymyxin B). In our case the standardized migration time of CST A in Figure 2A was 0.9924, in Figure 2B 0.9924, and in Figure 2C 0.9918. Therefore, the fluctuation of standardized migration time was negligible.

Changes that have been made:

No changes have been made.

Statement of the Reviewer 2:

In result section “Elevated levels of ALT and AST are often indicators of liver damage or disease, which is common when multiple antibiotics are used” give reference of the drugs used regarding liver damage (toxicity). Generalized statements are discouraged in scientific manuscripts.

Reply to the comments of the Reviewer 2:

Thank you for your comment. This statement is corrected and modified according to your comment. Reference [49] was added to support the statement.

Changes that have been made:

  1. Results and discussion

2.3. Lipidomics

… This correlates with the elevation of ALT and AST enzymes on the same day. The elevated levels of ALT and AST can be often indicators of temporary liver injury or damage, which is common with antibiotic use [77]. Liver dysfunction associated with an increase in ALT and AST...

References

  1. Doß, S.; Blessing, C.; Haller, K.; Richter, G.; Sauer, M. Influence of Antibiotics on Functionality and Viability of Liver Cells In Vitro. Curr. Issues Mol. Biol. 2022, 44, 4639-4657. https://doi.org/10.3390/cimb44100317.

Statement of the Reviewer 2:

Table 2 is showing the gradient elution; 15.50 to 15.60 min is very short time for gradient elution.

Reply to the comments of the Reviewer 2:

Thank you for your comment. It is true that such a step is very short in the LC gradient, but at this time (15.50 min - 15.60 min) no more lipids eluted from the column (the last identified lipid had a retention time of 14.95 min). This step together with the subsequent (2.40 min long) step is essential and required to wash out the non-polar components present in the sample. Moreover, this analytical method is a part of a standard operation protocol used at a convenient experimental and clinical workplace in Slovakia and Czech Republic.

Changes that have been made:

No changes have been made.

Statement of the Reviewer 2:

Comments on the Quality of English Language:

The language is not appropriate for a scientific manuscript. 

Reply to the comments of the Reviewer 2:

Thank you very much for your suggestion. According to the recommendation of the reviewer the quality of English language was improved by the use of common tools such as Writefull and Grammarly. Moreover, the revised manuscript was checked and modified by an English native speaker. 

Changes that have been made:

Extensive changes have been made through the whole manuscript. All changes in the manuscript are highlighted in yellow.

Round 2

Reviewer 1 Report

Comments and Suggestions for Authors

All my previous concerns have been addressed.

Author Response

The authors thank for you for your interest in our work and also for your kind decision.

Reviewer 2 Report

Comments and Suggestions for Authors

On page 9, line 363,the authors have stated "showed discrepancies in comparison with the findings by Mecatti et al. [64]" Can the authors give the reason for discrepancies. 

On page 9, line 342 "Therefore, the factor of different food intake will play an insignificant role here." Better to remove this sentence.

Better to present the values as creatinine clearance because in clinical settings dose adjustments are based on creatinine clearance not on serum creatinine or authors should provide the rationale of using serum creatinine instead of creatinine clearance. 

In results section, the authors have repeatedly use the term "significant" e.g. page 8, line 325 "Interestingly, we observed a significant drop in the concentration of" the significance need to be defined in methodology in statistics part.

Comments on the Quality of English Language

Minor changes
